# Sudden changes in nitrogen dioxide emissions over Greece due to lockdown after the outbreak of COVID-19

**Maria-Elissavet Koukouli[1]\*, Ioanna Skoulidou[1], Andreas Karavias[2], Isaak Parcharidis[2], Dimitris Balis[1], Astrid Manders[3], Arjo Segers[3], Henk Eskes[4] and Jos van Geffen[4]**

[1] Laboratory of Atmospheric Physics, Aristotle University of Thessaloniki, Greece.
[2] Department of Geography, Harokopio University, Athens, Greece.
[3] TNO, Climate, Air and Sustainability, Utrecht, The Netherlands.
[4] Royal Netherlands Meteorological Institute (KNMI), De Bilt, The Netherlands.
\* Correspondence: mariliza@auth.gr

**Abstract:** The unprecedented order, in modern peaceful times, for near-total lockdown of the Greek population, as means of protection against the Severe Acute Respiratory Syndrome CoronaVirus-2, commonly known as COVID-19, infection, has brought unintentional positive side-effects to the country's air quality levels. S5P/TROPOMI monthly mean tropospheric nitrogen dioxide ($NO_2$) observations show an average change of -34% to +20% [-39% to -5%] with an average decrease of -15% [-11%] for March and April 2020 respectively, compared to the previous year, over the six larger Greek metropolitan areas, attributable mostly to vehicular emission reductions. For the capital city of Athens, weekly analysis was statistically possible for the S5P/TROPOMI observations which revealed a marked decline in $NO_2$ load between -8% and -43% for seven of the eight weeks studied, in agreement to the equivalent OMI/Aura observations as well as the ground-based estimates of a Multi-Axis Differential Optical Absorption Spectroscopy ground-based instrument. Chemical transport modelling of the $NO_2$ columns, provided by the LOTOS-EUROS Chemical Transport Model, shows that the magnitude of these reductions cannot solely be attributed to the difference in meteorological factors affecting $NO_2$ levels during March and April 2020 and the equivalent time periods of the previous year. Taking this factor into account, the resulting decline was estimated to range between ~ -25% and -65% for the five of the eight weeks studied, with the remaining three weeks showing a positive average of ~ 10% which is postulated to the uncertainty of this methodology which is based on differences. As a result this analysis, we conclude that the effect of the COVID-19 lockdown and restriction in transport emissions over Greece is ~ -10%. As transport is the second largest, after the industrial, sector, that affects Greece's air quality, this occasion may well help policy makers in enforcing more targeted measures to aid Greece in further reducing emissions according to international air quality standards.

**Keywords:** Air quality; nitrogen dioxide; NOx; emissions; Sentinel-5P; TROPOMI; LOTOS-EUROS; COVID-19; pandemic; Athens; Greece

## 1. Introduction

In this work we aim to quantify the decline in tropospheric nitrogen dioxide ($NO_2$) levels over Greece during the ongoing Severe Acute Respiratory Syndrome CoronaVirus-2, commonly known as *COVID-19*, pandemic, as sensed by the space-borne S5P/TROPOMI, hereafter TROPOMI, instrument. By comparing the relative levels for the months of March and April for years 2020 and 2019, while properly accounting for the differences in meteorology using the simulations of a Chemical Transport Model, CTM, we quantify the improvement in local and regional air quality due to the reduced nitrogen oxides (NOx) emissions.

In the following sections, we provide basic information on tropospheric NOx, we focus on current knowledge of the nominal NOx emissions over Greece, we then present a brief overview of the capabilities of current and past satellite instruments in sensing abrupt atmospheric content changes and furthermore provide the dates when the different lockdown measures were enforced nationwide in Greece.

### 1.1. Nitrogen oxides in the troposphere

Nitrogen dioxide ($NO_2$) and nitrogen oxide (NO), referred to more commonly as nitrogen oxides (NOx), are important trace gases in the Earth's troposphere. NOx are emitted as a result of both anthropogenic activities, such as fossil fuel combustion and biomass burning, and natural processes, such as microbiological processes in soils, wildfires and lightning. In the presence of sunlight, the photochemical cycle of tropospheric ozone ($O_3$) converts

NO into $NO_2$ on a timescale of minutes and so $NO_2$ is considered a robust measure for concentrations of nitrogen
oxides (Jacob, 1999). For typical levels of the OH radical, the lifetime of NOx in the lower troposphere is less than
a day, normally a few hours depending on the season and the rates of the photochemical reactions [see for e.g.
Beirle et al., 2011; Mijling and van der A, 2012]. As a result, it is well accepted that $NO_2$ fluxes will remain relatively
close to their source which, first of all, makes it possible for NOx emissions to be well detected from space [see for
e.g. Stavrakou et al., 2008; Lamsal et al., 2010; van der A et al., 2008] but also precludes any transboundary pollution
effects which might otherwise hinder this study.

In the troposphere, $NO_2$ plays a key role in air quality issues, as it directly affects human health [WHO, 2016].
In the European Union, the evidence of $NO_2$ health effects has led to the establishment of air quality standards for
the protection of human health. Limit values for $NO_2$ are set at $200\,\mu g\,m^{-3}$ for 1 h average concentrations (with 18
exceedances permitted per year), and $40\,\mu g\,m^{-3}$ for annual average concentrations (European Council Directive
2008/50/EC, 2008). Concentrations above the annual limit value for $NO_2$ are still widely registered across Europe,
even if concentrations and exposures continue to decrease [EEA, 2019]. In Greece in particular, the annual average
standard of $40\,\mu g\,m^{-3}$ has not been exceeded between years 2007 and 2017 when assuming all in situ stations; the
traffic stations of Athens and Thessaloniki however show annual levels up to $45\,\mu g\,m^{-3}$ for years 2015 to 2017. It
hence follows logically that monitoring closely abrupt changes in NOx emissions for diverse locations plays a key
role in shaping future environmental policies and directives.

## 1.2. Nitrogen dioxide emissions over Greece

According to the EEA Report No 8/2019, updated by the EU 2019 Environmental Implementation Review for
Greece [EU, 2019], the country's NOx emissions by sector originate from road transport, industry (which mainly
covers the energy production and distribution sector), non-road transport, household and agriculture. The relative
percentages for NOx air emissions, separated by sector, as extracted from the 2018 Air Emission Account 2015
report by the Hellenic Statistical Authority [HAS, 2017], are: industry 48%, transport 22%, energy supply 18%,
manufacturing 6%, central heating 4%, agriculture 1% and others 1%. Based on the European Environmental
Agency, EEA, European Pollutant Release and Transfer Register, 77% of the reported industrial NOx/$NO_2$
emissions over Greece came from thermal power stations and other combustion installations. The monthly energy
balance reports, composed by the Independent Power Transmission Operator of the Hellenic Electricity
Transmission System (IPTO, 2020), show that the total energy requested for March 2020 [4.152GWh] was lower by
77  -2.1% than 2019 [4.224GWh] , whereas for April 2020 [3.527GWh] was -9.8% lower than 2019 [3.527]. These
reductions are quite typical of the seasonality of the energy consumption in Greece which peaks in December and
January, due to heating needs, and in July and August, due to cooling needs, with seasonal lows in spring (April
and May) and autumn (October and November). Furthermore, in Fameli and Asimakopoulos, 2016, it is reported
that the annual mean NOx emissions for Greece for years 2006 to 2012 can be attributed as follows, in order of
relevance: industry, 45±3%, road transport, 35±8%, shipping 11±3%, non-road transport, 10±4%, central heating,
5±2%, with agriculture and aviation showing an average of around 1± each. If we assume that years 2019 and 2020
were not exceptional in their temperature levels for the spring months, then it follows that changes in central
heating emissions will not be a significant part of the emission changes observed.

## 1.3. Sensing abrupt emission changes from space-borne sensors

Abrupt emission changes have already been reported using space-borne observations for a number of recent
local and continental circumstances. Castellanos and Boersma, 2012, reported significant reductions in nitrogen
oxides over Europe driven by environmental policy and the economic recession based on OMI/Aura observations
between 2004 and 2010. Vrekoussis et al., 2013 and Zyrichidou et al., 2019, report strong correlations between
pollutant levels and economic indicators showing that the 2008 economic recession has resulted in proportionally
lower levels of pollutants in large parts of Greece. The latter, for years 2008 to 2015, showed surprisingly that, while
the wintertime tropospheric $NO_2$ trends were negative, significant positive formaldehyde trends were observed
from space, shown to be due to increased usage of affordable indoor heating methods (e.g. fireplaces and wood
stoves). Space-sensed reductions in emissions, on a shorter time scale, have also been attributed to strict measures
enforced for benign reasons. Mijling et al., 2009, calculated, using OMI/Aura and CTM results, reductions in $NO_2$
concentrations of approximately 60% above Beijing during the 2008 Olympic and Paralympic Games. Ding et al.,

2015, showed a ~30% decrease in OMI/Aura columns, which was translated into a ~25% in actual emission levels during the Nanjing 2014 Youth Olympic Games.

Numerous first reports suggesting an improved air quality in after the COVID-19 lockdown was enforced have already been seen in major media outlets. Here we note the findings of Liu et al., 2020, who report for China, based on both OMI/Aura and TROPOMI, a 48% drop in tropospheric $NO_2$ from the 20 days averaged before the 2020 Lunar New Year to the 20 days after, which is 20% larger than that from recent years, and relate the increase in decline to the date of each Chinese province lockdown. Similar levels of tropospheric $NO_2$ decrease over different Chinese provinces are reported by Ding et al., 2020 and Miyazaki et al., 2020. Bauwens et al., 2020, based on the same sensors, also report an average $NO_2$ column drop over all Chinese cities of -40% relative to the same period in 2019, while the decreases in Western Europe and the U.S. were found to range between -20 to -38%. Goldberg et al., 2020, analyzed TROPOMI observations around large U.S. cities focusing on the effect of meteorological factors and report that meteorological variations between years 2019 and 2020 can cause columnar $NO_2$ differences of ~15% over monthly timescales. Compensating for meteorology, they then calculated a decrease in $NO_2$ levels between 9.2% and 43.4% among 20 cities in North America, with a median of 21.6%. Cersosimo et al., 2020, regridded the TROPOMI observations down to the 1x1km level and found that the reductions measured by air quality in situ measurements in the Po Valley, Italy, were very well reproduced by the satellite observations, while comparisons over less polluted regions in the South of Italy provide mixed results, which may also be attributed to the lower space-sensed levels. Vîrghileanu et al., 2020, also analysed the lockdown effect on Europe-wide pollution and report correlations to in situ observations ranging between 0.5 and 0.75, while also demonstrating the usefulness of such high spatial resolution satellite observations when used in tandem to other economic factors.

### 1.4.  The COVID-19 situation over Greece

A short review on the COVID-19 situation over Greece is given here, mainly focusing on providing the dates in March 2020 of the successive restrictive measures that affected NOx emissions which were placed nationwide. The country's General Secretariat for Civil Protection reacted quickly to the emerging situation in the neighbouring country Italy and long before the first causalities were reported, major festivities for the Carnival season planned for the 28th of February to the 2nd of March were cancelled, followed by cancellation of all other cultural and sporting activities on March 8th. On the 11th of March, all levels of education were suspended, when also a first wave of workplace closures begun and culminated on Monday 16th when all restaurants, coffee shops with sitting facilities, and in general the food [apart from supermarkets] and hospitality industry were shut down. In the following two days, all remaining retail activities were suspended apart from pharmacies. On Monday 16th restrictions on the size of public gatherings were announced and the public transport section [buses, trams, underground, trains] started to reduce capacity. On Monday 23rd of March, full restrictions on the people's movements were imposed with strict stay-at-home mandates, with exceptions for essential working personnel, including all religious-related congregations, with a complete and without exception restriction around the Greek Orthodox Easter holidays of the 19th of April. The country remained in full lockdown mode until May 4th. We should note here that most industrial activities continued normal operations, albeit with a skeleton crew, which might account for some of the higher load observed around the city of Athens where most of these activities are located.

### 2.   Materials and Methods

In this section we introduce the TROPOMI tropospheric $NO_2$ observations, the CTM LOTOS-EUROS simulations and the proposed methodology to account for the different meteorological conditions between the nominal period of March-April 2019 and the disrupted one of March-April 2020.

### 2.1.  TROPOMI NO₂ observations.

The recently launched TROPOMI instrument on the Sentinel-5 Precursor (S5P) mission [Veefkind et al., 2012] has been providing global atmospheric observations since early 2018. Its very high spatial resolution of $3.5 \times 7$ km², upgraded to $3.5 \times 5.5$ km² in August 2019, and improved signal-to-noise ratio compared to previous space-borne instruments, permits the detection of tropospheric pollution from small-scale emission sources and the estimation of very localized emissions from anthropogenic activities, such as industrial point sources, as well as regional fires.

Lorente et al., 2019, have already reported updated emissions over the Paris metropolitan area using TROPOMI observations, while Ialogno et al., 2020, have assessed the capabilities of this instrument in evaluating city-wide air quality levels compared to the more traditional ground-based and in situ $NO_2$ monitoring methods.

In this work we use the publicly available TROPOMI offline v1.2 and v1.3 for March-April 2019 and for March-April 2020 tropospheric $NO_2$ data accessed via the Copernicus Open Data Access Hub. The algorithm producing this data is described by van Geffen et al. (2019) and is based on the approach used for processing OMI/Aura $NO_2$ data within DOMINO and the FP7 Quality Assurance for Essential Climate Variables, projects (Boersma et al, 2011; 2018). Routine validation is being carried out by the Validation Data Analysis Facility, who also provide the Validation Report of the Copernicus Sentinel-5 Precursor Operational Data Products quarterly. The S5P tropospheric $NO_2$ columns are routinely compared to ground-based column data at 19 ground-based Multi-Axis Differential Optical Absorption Spectroscopy, MAXDOAS, stations. The latest report reports a negative bias of typically -22% to -37% for clean and slightly polluted conditions, reaching values of -51% over highly polluted areas (ROCVR #08, 2020). Furthermore, within ROCVR #08, the case of Athens was used as an example of the lockdown effects on $NO_2$ observations from both ground and space. It is shown that both the Athens MAXDOAS instrument (operated by the Institute of Environmental Physics, University of Bremen) and the S5P observations observe a significant drop in $NO_2$ levels between March 3rd and March 13th, in line to the first nation-wide measures on March 10th, 2020, with further restrictions on later days (closure of business, ban on non-essential movement, see https://www.bloomberg.com/news/articles/2020-04-17/humbled-greeks-show-the-world-how-to-handle-the-virus-outbreak.) $NO_2$ columns over Athens remained consistently low for weeks after, while the MAXDOAS instrument overall is performing very well, with a mean difference to the TROPOMI observations of $1.40\pm3.50\times10^{15}$ molecules/cm² (median of $0.25\times10^{15}$ molecules/cm²) for the 385 coincident days of observations between 01.05.2018 and 28.11.2020 (https://mpc-vdaf-server.tropomi.eu/no2/no2-offl-maxdoas/athens). We should note at this point that since in this work the main findings refer to relative differences between different time periods will be examined, absolute differences to standard instruments do not affect our findings, as the stability of the TROPOMI datasets is assured.

For the purposes of this work, orbital files over Greece, between 19° and 30°E and 34° and 42°N, were gridded onto a 0.10x0.05° grid for different temporal scenarios. The data have been filtered, as recommended, using the quality flag indicator $\geq 75$ assuring the data under this flag is restricted to cloud-free (cloud radiance fraction $< 0.5$) and snow-ice free observations. An example figure is presented in Section 3.1 (Figure 1) where the major NOx emitting sectors around Greece are prominent, the capital city Athens and the second largest city Thessaloniki, in the North, as well as emissions by one of the two largest thermal power plants, in Ptolemaida, also in the North and an important thermal power plant in Northern Macedonia, near the Greek border, where such trans-border pollution transport is often visible. The domain also includes major emissions from known locations in the Turkish Asia Minor coast, both originating from cities and major power plant activities, as well as the major shipping track that emerges from the Strait of Bosporus in Istanbul moving SW towards Athens before turning Westwards towards the Mediterranean Sea.

## 2.2. LOTOS-EUROS CTM simulations.

The open source chemical transport model LOTOS—EUROS v2.2.001 is used for the purposes of this study to simulate $NO_2$ columns over the Greek domain for March and April for years 2019 and 2020. The CTM model, https://lotos-euros.tno.nl/, is originally aimed at air pollution studies and simulates gases ($O_3$, NOx, $SO_2$, etc) as well as aerosols (sulfate, nitrate, PM10, PM2.5 etc.) in the troposphere. The gas-phase chemistry of the model is a modified version of CBM-IV (Gery et al., 1989). The Isorropia II module (Fountoukis and Nenes, 2007) is used for the aerosol chemistry, while detailed information on the model and its activity can be found in Manders et al., 2017. LOTOS-EUROS is the national air quality model for the Netherlands (Vlemmix et al., 2015), and has been used for specific studies as well to investigate $NO_2$ values (Timmermans et al., 2011; Curier et al., 2012; 2014). LOTOS-EUROS also participates at the operational Copernicus Atmosphere Monitoring Service (CAMS) consisting one of the seven CTMs that provide the official CAMS ensemble air quality forecasting service, while its capabilities were demonstrated during the Monitoring Atmospheric Composition and Climate, MACC, and continued in the MACC-II (Monitoring Atmospheric Composition and Climate: Interim Implementation) European projects (Marécal et al., 2015). Vlemmix et al., 2015, compared LOTOS-EUROS $NO_2$ tropospheric columns with MAXDOAS,

measurements, and found a good agreement between the two datasets, with a correlation coefficient between the daily averaged columns equal to 0.72. Schaap et al., 2013, compared the LOTOS-EUROS $NO_2$ simulations with OMI/Aura retrievals and also showed that the model captures the $NO_2$ spatial distribution satisfactorily and is able to explain 91% of the OMI signal variation across Europe, while the systematic difference was attributed to the summer period.

In this work, LOTOS-EUROS $NO_2$ simulations over Greece as those presented and discussed thoroughly in Skoulidou et al., 2020 are used. The model uses off-line meteorology extracted from the Operational Forecast Data from the European Centre for Medium-range Weather Forecasts, ECMWF. These meteorological model level fields cover temperature, boundary layer height, specific humidity, wind components, half level pressures, cloud coverage, cloud liquid and ice water content, rain and snow water content, total cloud coverage, convective and large-scale precipitation, wind components at 10 meters, etc. In addition, surface fields that include orography, soil type, land/sea mask, sea surface and soil temperature, dew point temperature at 2 meters, surface latent and sensible heat fluxes, surface solar downward radiation, and similar, are also included. The different level type fields are obtained at a 3-hour temporal resolution while the surface fields are imported at a 1-hour resolution. The horizontal resolution of the meteorological input fields is 7km×7km. In the vertical domain, the model distinguishes 10 levels which extend from the surface to about 175hPa. The height of these levels refers to the levels of the ECMWF meteorological input data that are further used to drive the model runs. The initial and boundary conditions are constrained from a coarser run of LOTOS-EUROS that is performed over the larger European domain (15° W to 45° E and 30° – 60° N) with a resolution of 0.25°×0.25°, as discussed in Skoulidou et al., 2020. The anthropogenic emissions are provided by the CAMS-REG (CAMS Regional European emissions) inventory for the year 2015 with a horizontal resolution of 0.1°×0.05° (Kuenen et al., 2014).

The LOTOS-EUROS $NO_2$ simulations over Greece reproduce very well the spatial variability of the TROPOMI $NO_2$ columns over Greece, capturing the locations of low and high $NO_2$ columns (see their Figures 11 and 12 for the summer and winter period respectively). The spatial correlation between the simulations over Athens (Thessaloniki) and the TROPOMI observations is 0.95 (0.82) in summer and 0.82 (0.66) in winter, with a bias of ±18% (+4 to -27%). Furthermore, comparisons to MAXDOAS systems located in both cities have shown that LOTOS-EUROS simulates very well the diurnal variability of the $NO_2$, with biases between 0% and 30% depending on the season for the overpass time of TROPOMI [around 12:00 UTC] quite in agreement to what was found in Vlemmix et al., 2015 and Blechschmidt et al., 2020.

### 2.3. Comparative Methodology.

While it would make sense to simply compare the $NO_2$ levels over Greece for the two periods, assuming that the emission sources have not changed dramatically between 2019 and 2020, one should not discard the effects that various meteorological parameters have on $NO_2$ levels (Goldberg et al., 2020). Meteorological conditions, such as wind speed, temperature inversions and the depth of the boundary layer, often play pivotal roles in local air quality levels (Jacob and Winner, 2009). The ambient levels of secondary $NO_2$ pollution are determined through the accumulation or dispersion of pollutants, low or high solar irradiances, regional transport of clear or polluted air and atmospheric chemistry for the formation of secondary species, in this case via the chemical coupling of $NO_x$ with $O_3$ (for e.g. Seo et al., 2017).

To ensure that the observed decrease in $NO_2$ levels was not due to diverse meteorological conditions between one year and the next, relative differences on $NO_2$ columns provided by the LOTOS-EUROS model are calculated and their average magnitude is set as the expected contribution by the different meteorology. This forms a standard level above which we expect COVID-19 related, i.e. emission-related, reductions. The premise of this thinking is as follows: differences in the satellite observations will contain the intertwined effect of differences in meteorology on concentrations and of differences in emissions. For the model we keep the emissions constant for the two periods but use the meteorology of 2019 and 2020 so that we can isolate the impact of meteorology on concentrations. We cannot of course exclude the possibility that the LOTOS-EUROS model has biases in the resulting $NO_2$ column depending on the meteorological conditions. In Skoulidou et al., 2020, differences in night-time surface concentrations between in situ observations and model simulations were found which were attributed to modelling uncertainties in mixing under stable conditions. Within the methodology followed in this work we expect any possible biases to cancel out in the difference fields calculated.

In this point we should stress that the satellite observations are more often than not gap-ridden, since in the
suggested screening all but nearly clear-skies remain. Spring-time months are rainy months, even for typically
sunny Greece, which means that a one-to-one comparison of the satellite observations for the two periods, even on
a weekly basis, is usually impossible. During our analysis it was found that, for e.g. the last week of March of 2020,
the first week of full lockdown, was fully cloudy for the Northern Greece even though the equivalent week in
March 2019 was all sunny. As a result, weekly comparisons were only possible for the major $NO_2$ hotspot over
Greece, the city of Athens, while the rest of the domain was examined on a monthly basis.

In technical terms, the LOTOS-EUROS simulations were performed on the entire timeline as discussed in
Section 2.2 but were restricted, on a daily basis, to the TROPOMI pixels that actually provided an observation when
performing the temporal averaging and producing the comparative plots. Even though a direct comparison of the
CTM results to the satellite data is not the focus of this paper, we imposed this filter to make sure that the same
256  days with the same meteorological conditions were viewed by both methods. Furthermore, as discussed in Eskes
and Boersma, 2003, so as to properly compare the modelled and measured columnar data, we applied the
TROPOMI averaging kernels, AKs, to the modelled profiles before extracting the CTM columns. The LOTOS-
EUROS CTM includes a module that imports the TROPOMI $NO_2$ orbital files in a pre-defined format, performs all
the necessary filtering, regridding and averaging of the datasets and executes the AK convolution of the nearest in
time observation to the CTM $NO_2$ simulated profile before outputting the profile and columnar information on the
predefined spatiotemporal grid.

**3.    Results**

In the following section, we first show the effect on monthly $NO_2$ levels over the entire domain, the six Greek
cities with the largest number of inhabitants and we then present a more in depth analysis, on a weekly basis, for
the city of Athens, also examining the long term variability of tropospheric $NO_2$ levels over the capital city using
fifteen years of space-born observations by the Ozone Monitoring Instrument, OMI/Aura, as well as the air quality
*in situ* measurements of the Greek Ministry Environment & Energy network, reporting to the European
Environmental Agency.

**3.1.  Lockdown effects on monthly $NO_2$ levels**

In Figure 1 the monthly mean tropospheric $NO_2$ levels over Greece, the Northern neighboring countries, the
Aegean Sea as well as the coast of Turkey and the Istanbul area, are shown for year 2019 in the upper panel, year
2020 in the middle panel, their absolute difference in the lower panel, for the month of March in the left column
and the month of April in the right column. Even though the hotspots appear strong for year 2019, with discrete
shipping tracks and ground-tracks over Turkey showing clearly, the different meteorological conditions between
March and April obviously affect both the location of the maxima as well as the absolute level of those maxima.
Since Greece gradually entered full lockdown mode within the first three weeks of March, while Turkey imposed
intermittent movement restrictions from the beginning of April onwards, the $NO_2$ hotspot around the megacity of
Istanbul and the Bosporus Strait is still pronounced in March 2020 [upper left] while in Greece most of the smaller
urban emission points are missing, and Athens is shown in sharp decline. In April 2020, the Turkish hotspots are
also reduced in magnitude, as expected. In the following sections we focus on specific hotspot locations and
introduce numerical findings.

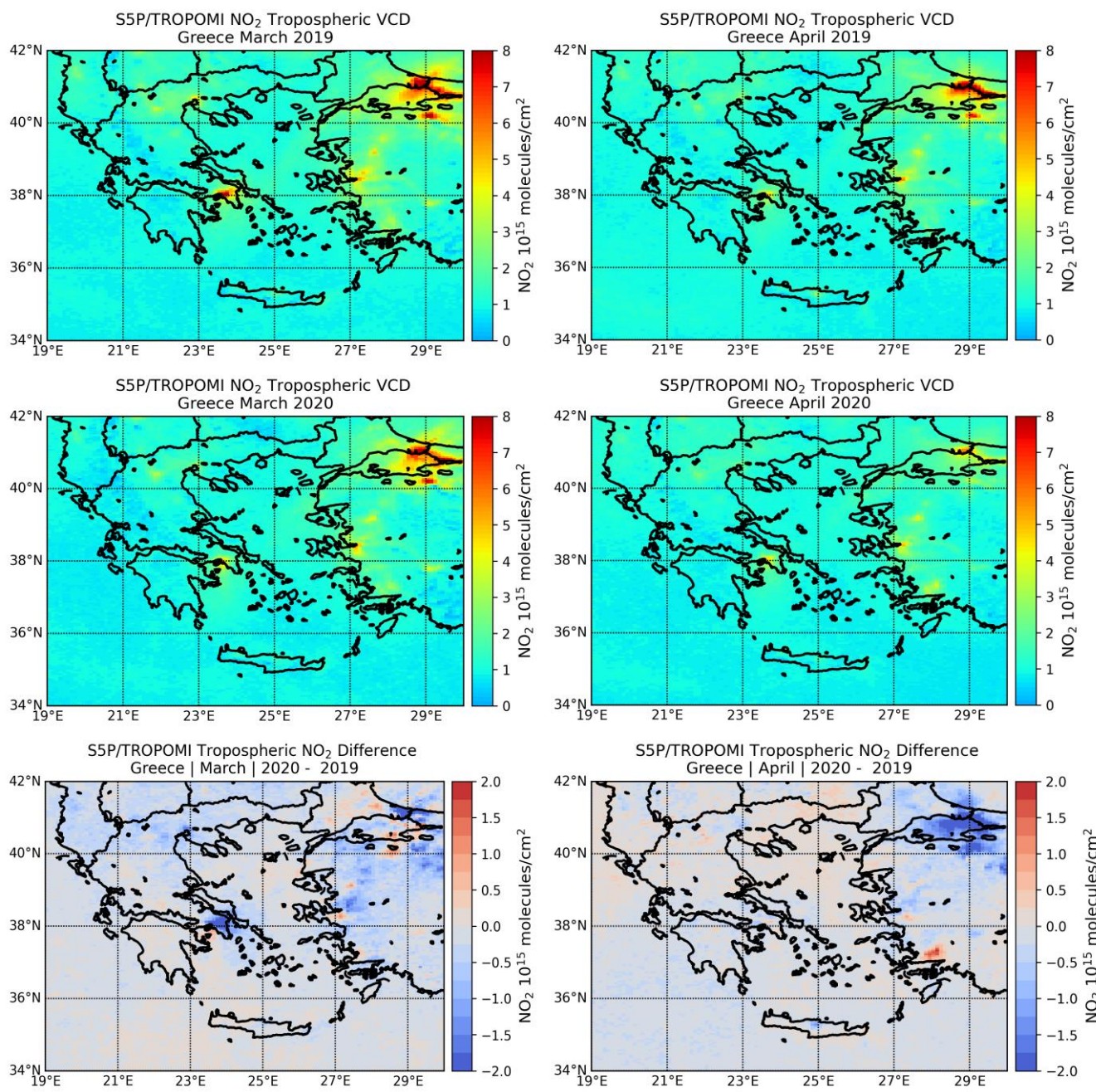

Figure 1 Monthly mean TROPOMI tropospheric NO₂ columns, in $10^{15}$ molecules/cm², for March [left] and April [right column] for the 2019 [top], 2020 [middle] and their absolute differences [lower panel].

In Figure 2 the monthly mean TROPOMI tropospheric NO₂ columns, in $10^{15}$ molecules/cm², are depicted for March 2019 [left] and 2020 [right] for six major cities in Greece top to bottom, namely, Athens [37.98° N, 23.72° E], Thessaloniki [40.64° N, 22.94° E], Larisa [39.63° N, 22.41° E], Volos [39.36° N, 22.95° E], Patra [38.24° N, 21.73° E] and Heraklio [35.33° N, 25.14° E]. We focus on the locations where major transport emissions are expected since.these six cities, according to the HAS, 2011, census, host 4.45 out of the 10.8 million of the Greek population (Table S1). Even though the NO₂ levels are low over the four smaller cities, we were interested in examining the ability of TROPOMI in sensing both the load and expected changes for these, relatively clean, cities [numeric results are given in Table 1]. The equivalent maps for April 2019 and 2020 are presented in Figure S1.

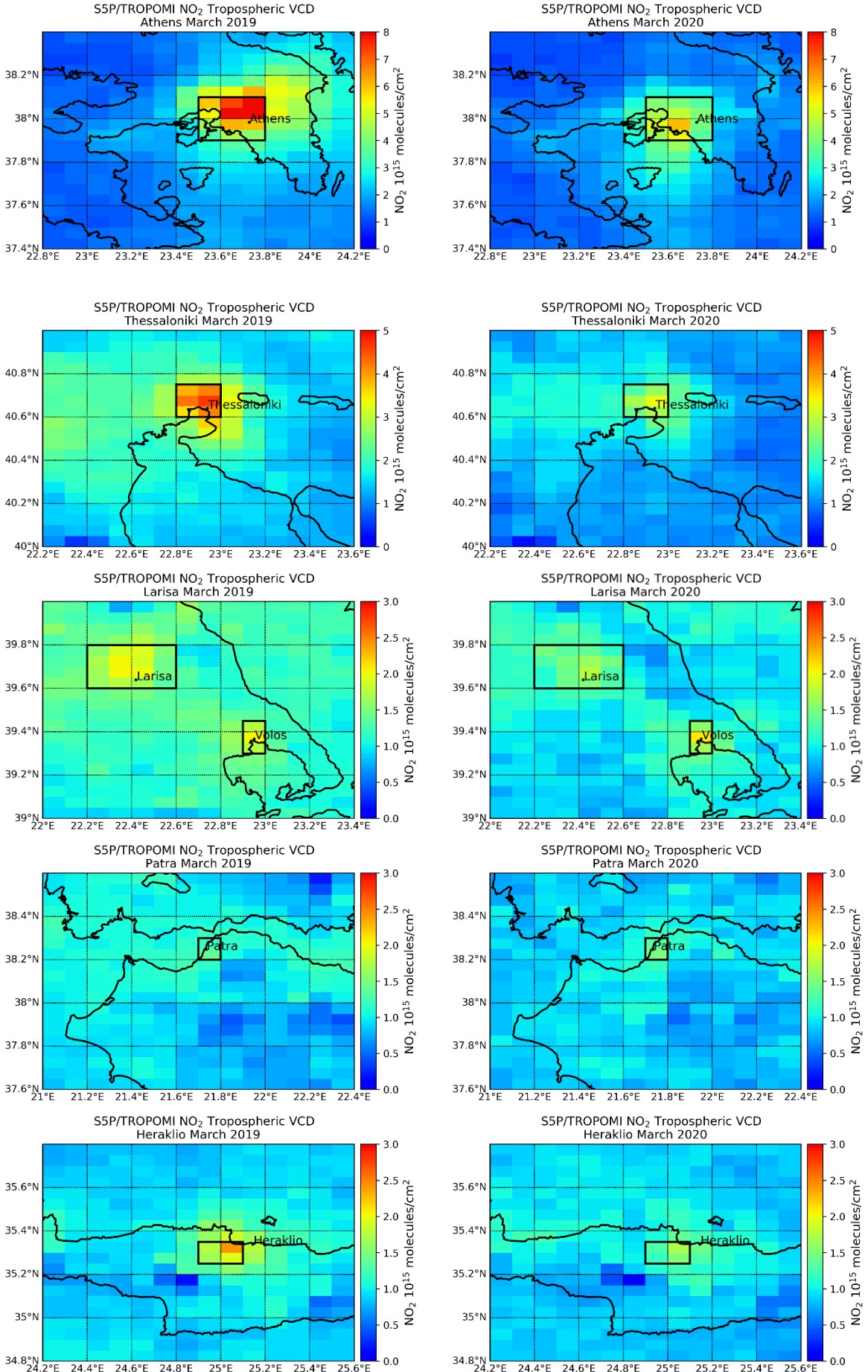

Figure 2. Monthly mean TROPOMI tropospheric NO₂ columns, in 10¹⁵ molecules/cm², for March 2019 [left] and March 2020 [right] for the five
major cities in Greece. First row, Athens; second, Thessaloniki; third, Larisa and Volos; fourth, Patras and fifth, Heraklio. The boxes mark the
pixels used in the numerical analysis.

In Figure 3 the monthly mean TROPOMI tropospheric NO₂ columns, in 10¹⁵ molecules/cm², for March [upper]
and April [lower] for the 2019 [blue] and 2020 [orange] are shown for the six major cities in Greece, from left to
right, Athens, Thessaloniki, Larisa, Volos, Patra and Heraklio. Overall, the NO₂ levels are higher in all cases for
both March months, than the equivalent April ones, and are proportional to the city population, with Athens and
Thessaloniki showing the highest levels while the remaining four present similar NO₂ loading conditions. It is
hence not surprising considering these rather low monthly mean reported satellite estimates, which approach the
level of detectability of the satellite sensor that the changes vary widely from one location to the next and not
always in the expected manner. We already note here that the associated standard deviation on the monthly mean
levels for the four smaller cities is quite large, and might affect the robustness of findings later on in this work. In
Table 1, the full statistics that relate to Figure 3 are given, where for the month of March, the relative differences in
NO₂ loading sensed by the satellite sensor between 2019 and 2020 are shown to range from -3 to -34% in all cases
except for the port city of Patra, where absolute changes of 0.12x10¹⁵ molecules/cm² result in percentage differences
of +20%. Similarly, for the month of April, relative changes range from -39% to +5% however these mostly result
from extremely small absolute changes of 0.06x10¹⁵ molecules/cm² (Athens) or 0.08x10¹⁵ molecules/cm² (Volos). The
equivalent bar plot for the CTM tropospheric NO₂ columns is given in Figure S2, in the same format as Figure 3,
and the relevant statistics in Table S2. We note here that, for the possible available observations per location, for
the month of March there were slightly less available pixels for year 2020 than 2019, on average ~-15% and range
between -5% and -22%, with the highest difference for Thessaloniki which was overcast the entire final week of
March 2020 as already discussed. For April, slightly more pixels are available for year 2020, on average +5% and
range between 1% and 11%.

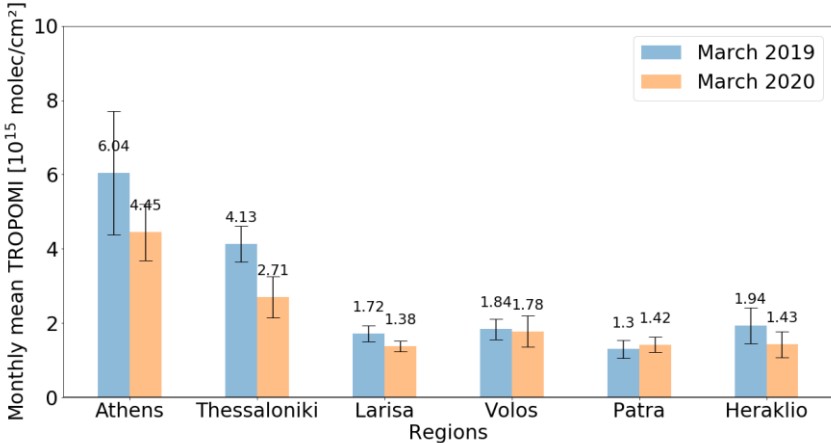

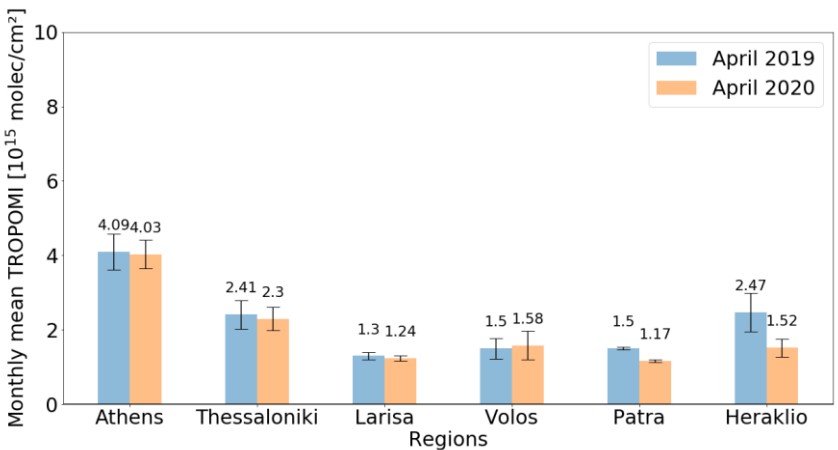

Figure 3. Monthly mean TROPOMI tropospheric $NO_2$ columns, in $10^{15}$ molecules/cm², for March [upper] and April [lower] for the 2019 [blue] and 2020 [orange] for the five major cities in Greece, from left to right, Athens, Thessaloniki, Larisa, Volos, Patra and Heraklio.

Table 1. Monthly mean TROPOMI $NO_2$ levels [$10^{15}$ molecules/cm²] over major cities in Greece for March [left block] and April [right block] for year 2019 and 2020 and their relative difference, standard deviation and number of pixels [in brackets].

| Location | 03.2019 | 03.2020 | % diff | 04.2019 | 04. 2020 | % diff |
|---|---|---|---|---|---|---|
| Athens [12] | 6.04±1.65 | 4.45±0.76 | -26% | 4.09±0.49 | 4.03±0.38 | -1% |
| Thessaloniki [6] | 4.13±0.34 | 2.71±0.38 | -34% | 2.41±0.28 | 2.30±0.22 | -5% |
| Larisa [16] | 1.72±0.22 | 1.38±0.17 | -19% | 1.30±0.13 | 1.24±0.09 | -5% |
| Volos [3] | 1.84±0.13 | 1.78±0.21 | -3% | 1.50±0.14 | 1.58±0.19 | +5% |
| Patra [2] | 1.30±0.10 | 1.42±0.19 | +20% | 1.50±0.02 | 1.17±0.01 | -18% |
| Heralkio [4] | 1.94±0.29 | 1.43±0.08 | -26% | 2.47±0.30 | 1.52±0.14 | -39% |

In Figure 4, upper, the monthly mean absolute differences in tropospheric $NO_2$ columns ($10^{15}$ molecules/cm²) between 2020 and 2019 are shown for TROPOMI [orange] and LOTOS-EUROS [blue] for the six major cities in Greece, for Athens, Thessaloniki, Larisa, Volos, Patra and Heraklio. We opted to show absolute differences here, and not percentage ones as might be expected, since a small relative change on a low $NO_2$ abundance would result in the erroneous message of a large reduction, as was already shown in Figure 3. In the lower panel of Figure 4, the emission changes are quantified in the following manner: the percentage differences for LOTOS-EUROS between 2019 and 2020 are calculated, as the equivalent ones seen by TROPOMI. By subtracting the two percentage differences, and not directly comparing the two, the actual $NO_2$ emission reduction may be quantified.

This percentage difference is found to be -16% for Athens, -12% for Thessaloniki, -34% for Larisa, -22% for Volos, -17% for Patra and +19%, for Heralkio. This study shows that, for relatively low tropospheric $NO_2$ columns of the order of $1.5 \times 10^{15}$ molecules/cm² this methodology which is based on differences, may result in unexpected numerical findings. Similar studies which have examined the effects of the COVID-19 lockdown on air quality based on satellite observations have focused on Eastern China, on specific US and Canadian locations, as well as over the Po Valley in Italy, which observe orders of magnitude higher tropospheric $NO_2$ columns even for the reduced emissions period. As a result, our main tentative finding is that over Greece, a ~-10% reduction in tropospheric $NO_2$ columns as sensed by the S5P/TROPOMI instrument may be attributed to the reduced emissions due to the COVID-19 pandemic. We hence continue this study focusing only over the location with the highest observed tropospheric $NO_2$ columns, the capital city of Athens, in a weekly temporal scale, so as to possible refine this estimate.

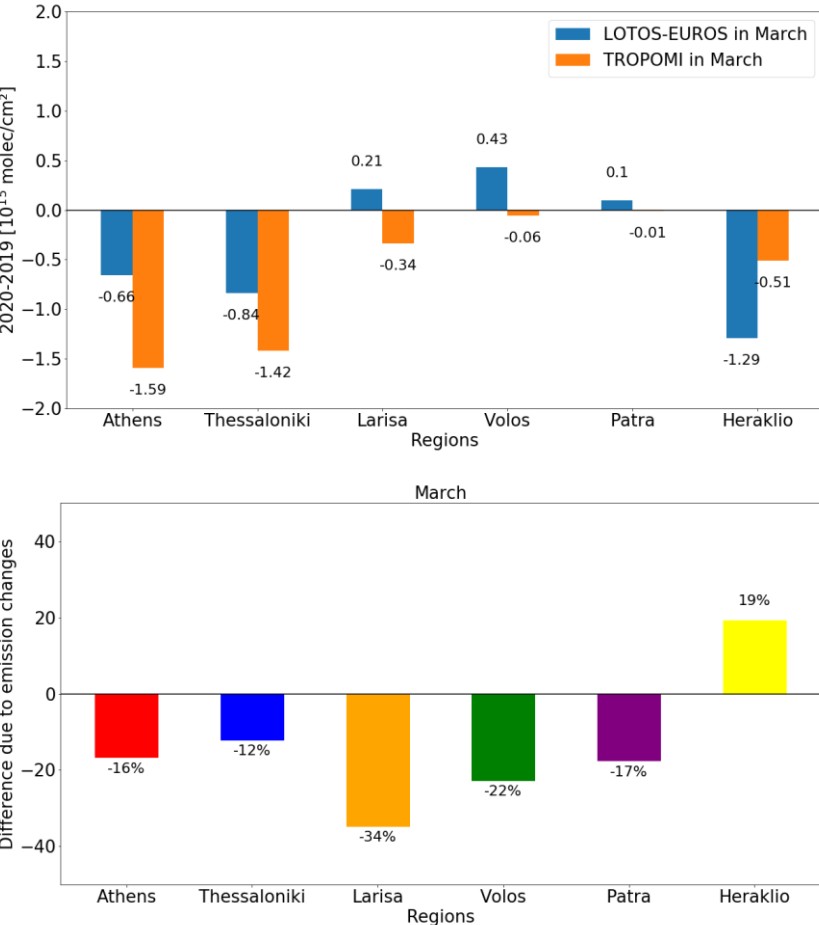

Figure 4. Upper. Monthly mean absolute differences in tropospheric NO₂ columns (10¹⁵ molecules/cm²) between 2020 and 2019 are shown for
TROPOMI [orange] and LOTOS-EUROS [blue] for the five major cities in Greece, from left to right, Athens, Thessaloniki, Larisa, Volos, Patra
and Heraklio. Lower. The percentage differences that may be attributable to emission changes.

**3.2. Lockdown effects on weekly NO₂ levels over Athens**

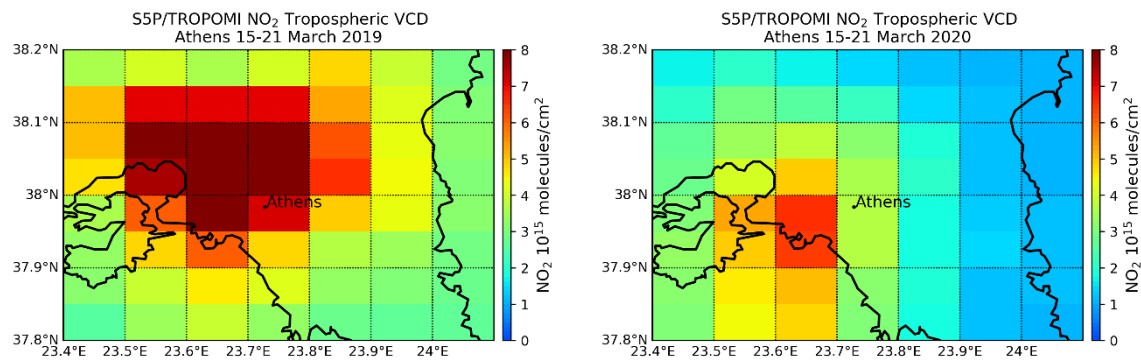

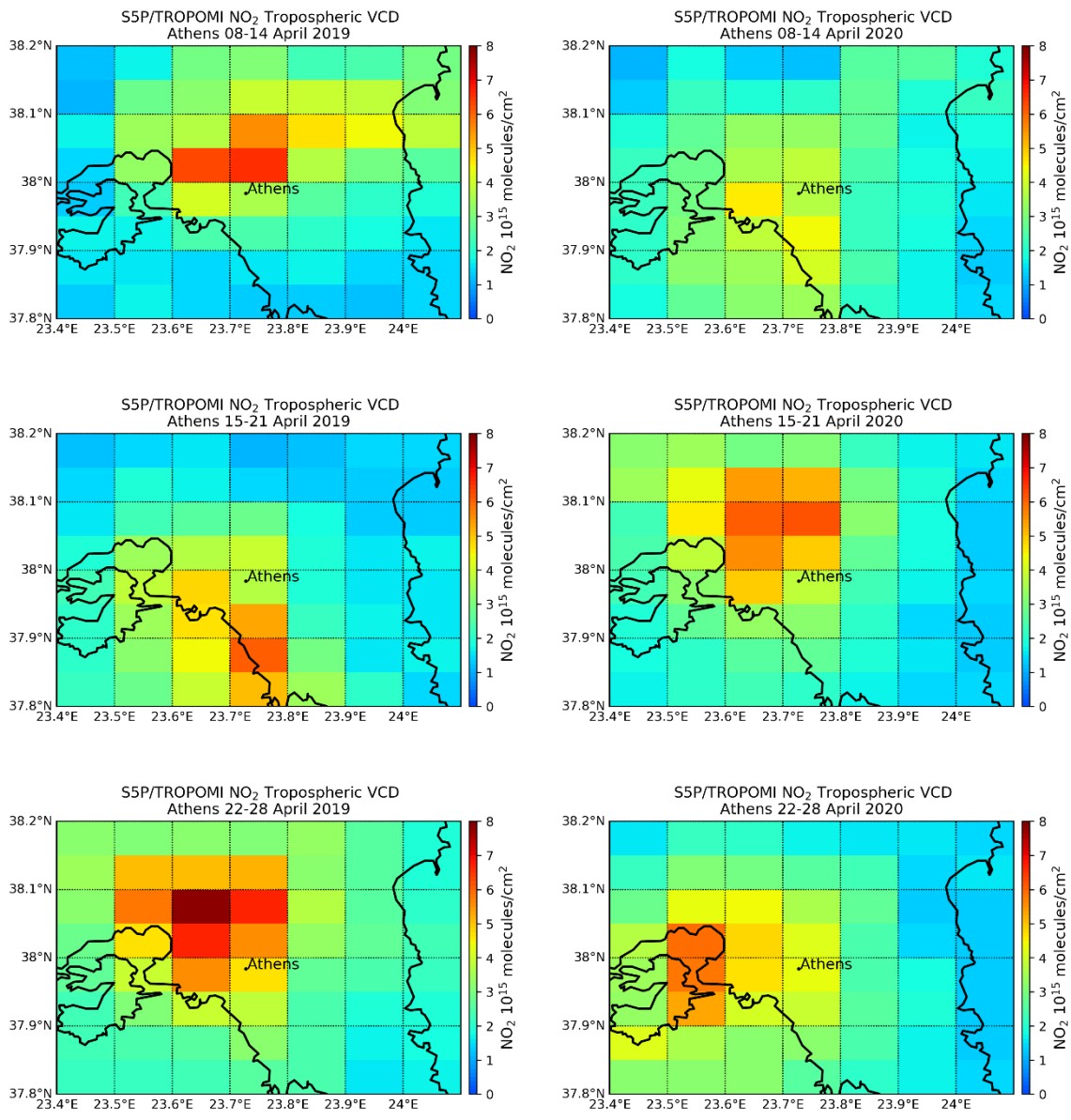

Figure 5. Weekly mean TROPOMI tropospheric NO$_2$ columns, in 10$^{15}$ molecules/cm$^2$, over Athens for 2019 [left] and 2020 [right]. First row, 15-21 March 2019; second, 8-14 April, third, 15-21 April and fourth, 22-28 April.

Without disregarding the possible contribution of central heating to total NOx emissions, the largest decrease due to the COVID-19 lockdown is indeed observed over the main Greek hotspot, the city of Athens and its surroundings. In Figure 5, weekly mean TROPOMI tropospheric NO$_2$ columns, in 10$^{15}$ molecules/cm$^2$, over Athens for 2019 [left] and 2020 [right] are shown for weeks 15-21 March (first row), 8-14 April (second row), 15-21 April (third row) and 22-28 April (fourth row). Apart from the obvious reduction in magnitude during the lockdown months, what is most prominent in this composite is the effect of the winds for both the location of the local maximum as well as the spread of the pollution plume, which further strengthens our decision not to perform one-on-one comparisons between the different NO$_2$ fields. In numbers, the average weekly NO$_2$ load over Athens sensed by TROPOMI is presented in Figure 6 where the 2019 averages are shown in blue and the 2020 ones in orange for weeks of March and April. Out of the 12 pixels considered for this sub-domain, which may provide up to 84 measurements for each week in the case of clear skies, for year 2019 an average of 53±16 [median of 52] clear sky S5P/TROPOMI observations where found whereas for year 2020 an average of 52±25 [median of 56]. Even

though the representativeness of the weekly levels can by no means be considered equal between the years, apart from the penultimate week, TROPOMI reports lower $NO_2$ columns ranging between -8% to -43%. The MAXDOAS observations over Athens also show a very similar behavior, reporting $6.77\pm6.85\times10^{15}$ and $3.60\pm1.83\times10^{15}$ molecules/cm$^2$ for March and April 2019, and $2.76\pm3.17$ and $2.77\pm2.44\times10^{15}$ molecules/cm$^2$ for March and April 2020 respectively, showing a much larger reduction for the month of March than the month of April.

The meteorology over these eight weeks over Athens shows that, temperature-wise, in 2019, the entire month of March as well as the first three weeks of April, had very similar levels with a very hot spell affecting the last week of April 2019 which was also Easter Week in Greece. In 2020, a cold front appeared during the third week of March which lasted until mid-April when warmer weather appeared and remained (Figure S5). The mean vector wind speed and direction, overlain as arrows in Figure S5, is very similar with mostly predominant northern winds and very few cases of southerly winds. In the equivalent rose diagrams, Figure S6, we note again that the main wind directions appear similar between the two periods [2019 in the left and 2020 in the right column] apart from the last week of April [bottom row] were indeed the two weeks had very different directions, for the same magnitude. Note that the percentiles are not constant between rose diagrams.

A question that often arises when examining a relatively short time period of a highly varying atmospheric species is whether the period considered as "normal" was indeed normal from a climatological point of view. We have opted to create a climatological mean based on the daily OMI/Aura $NO_2$ cloud-screened tropospheric column L3 global gridded 0.25x0.25° v003 product (Krotkov et al., 2017; 2019) accessed from the NASA EarthData Giovanni repository. The monthly variability of the tropospheric $NO_2$ load over Athens for year 2019, in blue, and 2020, in orange, is shown in Figure 7, upper panel, as the percentage difference from the climatological mean (grey shaded area). As is also observed by the TROPOMI instrument, the OMI observations also reveal a higher decrease for March 2020 compared to March 2019, than the equivalent decrease for the April months. In Figure 7, lower panel, the weekly variability of the tropospheric $NO_2$ load over Athens for year 2019, in blue, and 2020, in orange, is shown, starting in week 1, the first week of March and covering eight weeks. In this representation as well, it is shown that the weeks of March of 2020 were further from the climatological mean compared to the April 2020 ones, while the April 2019 weeks show overall lower $NO_2$ loads, as also shown by TROPOMI (Figure 6).

Another question that is also often discussed when examining such abrupt changes in localized emission sources is whether in situ surface measurements depict the changes in the same order and magnitude. For the case of the COVID-19 pandemic, a number of studies for European locations have appeared with surprising findings; in Ropkins and Tate, 2021, measurements from automated monitoring stations across the UK showed abrupt $NO_2$ decreases at the onset of the UK lockdown, between ~25 to 50% at urban traffic and urban background stations. Surprisingly, after the initial abrupt reduction, gradual increases were then observed through the rest of the UK lockdown period. A similar finding is reported by Dakre, Mortimer and Neal, 2020, who show that the in situ air quality stations in the north and middle of England measure a decrease in $NO_2$ concentrations in the lockdown period of 17/03/20 – 30/04/20 whereas stations in the south of England measured an increase in $NO_2$ concentrations. Putaud et al., 2020, studied in situ concentrations from an urban background and a regional background station in the North of Italy and showed that that $NO_2$ concentrations decreased as a consequence of the lockdown by -30% and -40% on average, respectively.

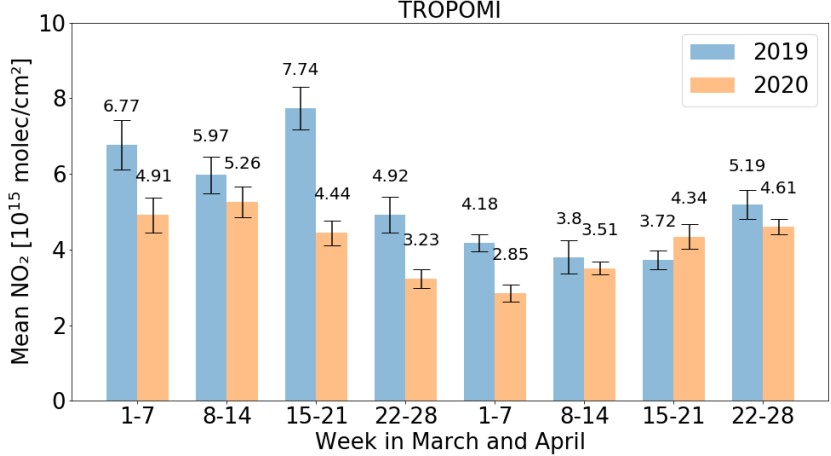

Figure 6. Weekly mean TROPOMI tropospheric NO₂ columns, in 10¹⁵ molecules/cm², for weeks in the 2019 [blue] and 2020 [orange] for Athens.

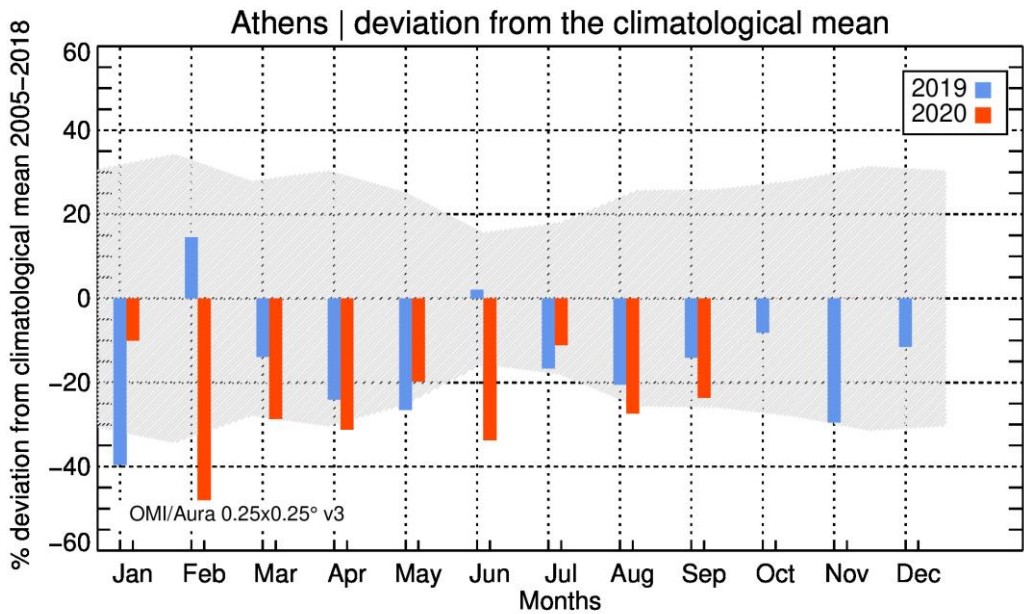

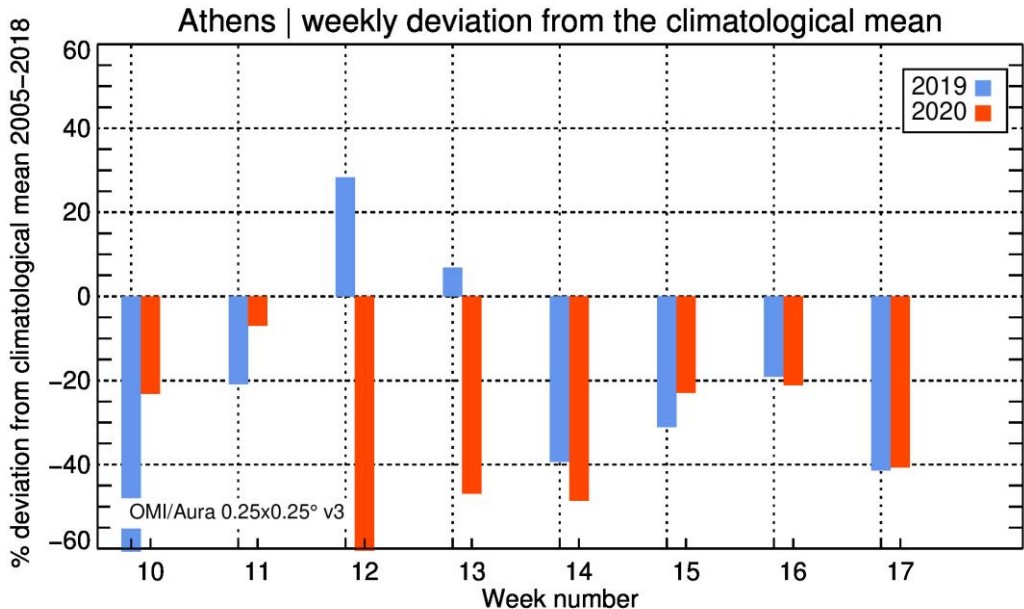

Figure 7. Upper. The OMI/Aura v003 L3 gridded cloud-screened tropospheric NO₂ monthly deviations from the climatological mean (grey shaded area) for year 2019 (in blue) and year 2020 (in orange.) Lower. The OMI/Aura v003 L3 gridded cloud-screened tropospheric NO₂ weekly deviations from the climatological mean for year 2019 (in blue) and year 2020 (in orange) starting in week 10, the first week of March.

For the purposes of this discussion we have analyzed the in situ surface NO₂ measurements reported by seven air quality stations around Athens, with their locations shown in Figure S3, and their individual monthly mean variability for 12:00 UTC is shown in Figure S4. These long-term observations are maintained by the Greek Ministry Environment & Energy, YPEKA, network who further report them to the European Environmental Agency, EEA, and are officially designated as industrial, urban and suburban locations. The measurement time closest to the TROPOMI overpass time over Athens was chosen to calculate a climatology between years 2005 and 2018, and in Figure 8, upper, the monthly mean NO₂ levels calculated from the time series shown in Figure S4, is presented The grey line and shaded area show the seasonal variability of the mean surface concentrations, in μgrams/m³, with higher levels during wintertime months and lower during summertime. Similar NO₂ concentrations are reported for both months of March 2019 (in blue) and 2020 (in purple), at the lower statistical level, while an unexpected increase for April (and May) 2019 show large differences to the lows found in April 2020, the full COVID lockdown month for Greece. This finding is clearer in the Figure 8, lower panel, where the monthly deviation of year 2019 (in blue) and year 2020 (in orange) are given as bars, overlaid against the grey shaded area which shows the variability of the climatological means. Contrary hence to what the space-born observations by both TROPOMI and OMI as well as the ground-based MAXDAS measurements show, the in situ measurements report a similar difference similar difference to the climatological mean for the months of March while April 2019 appears in the positive range, while April 2020 beyond the lower statistical level. Grivas et al., 2020, compared climatological hourly NO₂ concentrations measured by an urban background station in the Athens basin (not included in our work) for years 2016-2020 to days corresponding to the pre-lockdown (March 1-22), lockdown (March 23 – May 10) and post lockdown (May 11-31) periods of 2020. Overall, they report -6%, -41.5% and +8.7% between the periods of year 2020 to the 2016-2020 equivalent.

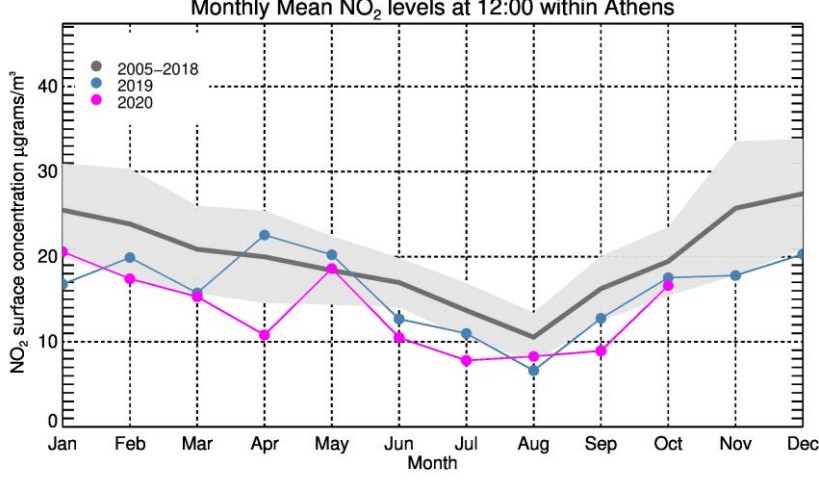

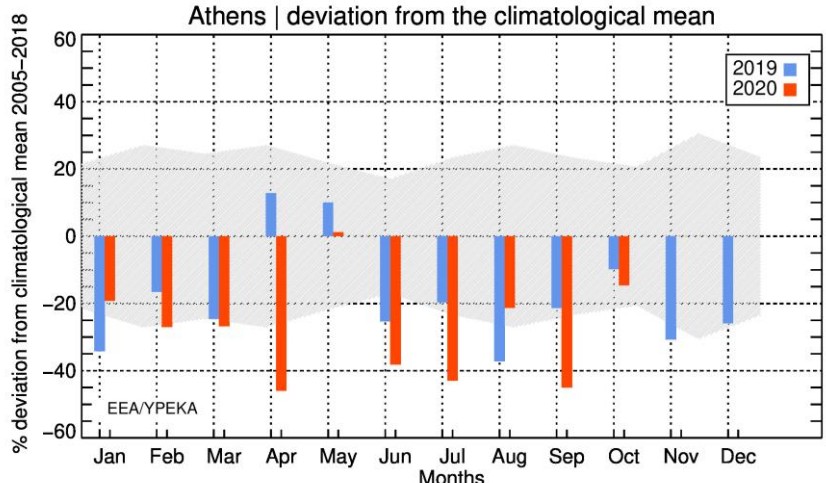

Figure 8. Upper. Monthly mean $NO_2$ surface concentrations, in μgrams/m³, for the climatological mean of 2005 to 2018 shown as a dark grey line and standard deviation (grey shaded area), year 2019 (in blue) and year 2020 (in purple) calculated from the levels reported by seven air quality stations, as shown in Figure S4. Lower. The monthly percentage deviation from the climatological mean (shaded grey area) for the months of 2019 in blue and 2020 in orange.

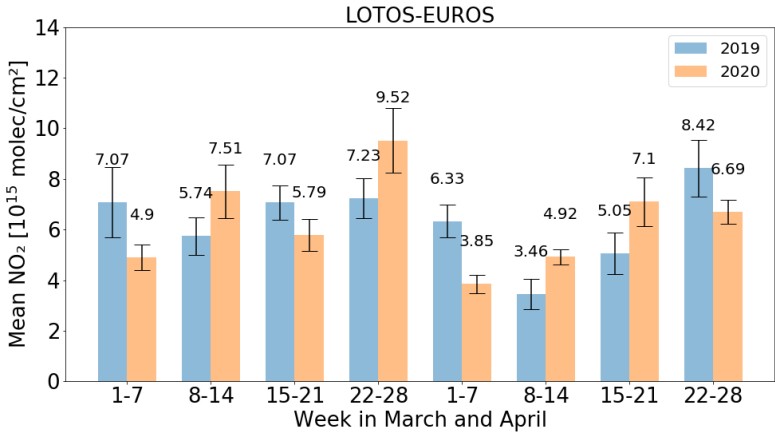

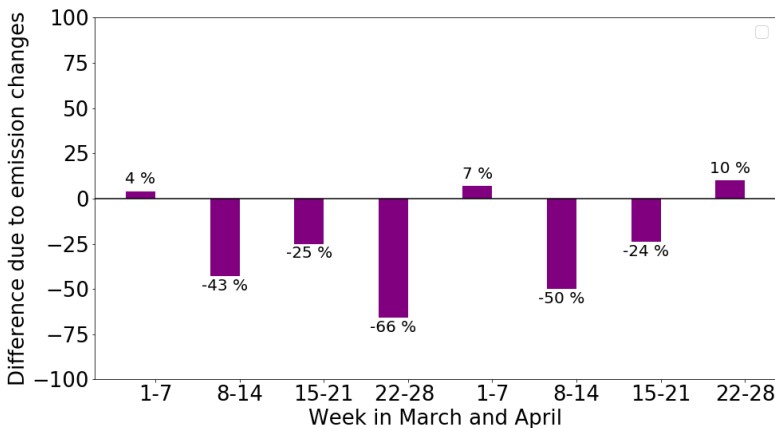

Figure 9. Upper. Weekly mean LOTOS-EUROS tropospheric NO₂ columns, in $10^{15}$ molecules/cm$^2$, for weeks in the 2019 [blue] and 2020 [orange] for Athens. Lower. The percentage differences attributed to emission changes, revealing the actual magnitude of the NOx emissions decrease.

The contribution of the meteorological factors to the observed tropospheric NO₂ load can be assessed by the equivalent LOTOS-EUROS weekly averages, shown in the upper panel of Figure 9. As for Figure 4, bottom, the percentage difference of the LOTOS-EUROS simulations between 2019 and 2020 are calculated, as were those for the TROPOMI equivalent weekly means (Figure 6). The difference between those two relative differences is given in the lower panel of Figure 9. The fact that the CTM predicted an increase in NO₂ production for most weeks, under the assumption that the primary emissions remained stable between the two years, results in higher reduction levels ranging between -24% and -66% for 5 of the 8 weeks studied, while an assumed increase in emissions is calculated for the remaining 3 weeks with levels between +4 and +10%. These increases in emission levels, which are not corroborated by the in situ observations, give us an estimate on the uncertainty of this methodology, at ~10%. Even so, the average difference in emissions over Athens for these eight weeks is calculated at ~-20% from the S5P/TROPOMI tropospheric NO₂ observations.

## 4. Conclusions

In this work, Sentinel-5P/TROPOMI tropospheric NO₂ observations were studied in order to examine the possible positive effect on Greek air quality caused the recent COVID-19 pandemic lockdown. The country enforced severe movement restrictions and entire economic sectors gradually were shut down, starting from the

last weekend of February and gradually, activity per activity, reaching a total lockdown in effect from Monday 23rd up to May 4th. The time period between March and April 2020, and the equivalent weeks in 2019, were analyzed and compared for six, largest in population, cities in Greece on a monthly basis. TROPOMI monthly mean tropospheric nitrogen dioxide, $NO_2$, observations showed a change of between -34% and +20% [-39% to +5%] with an average of -15% [-11%] for March and April 2020 respectively, compared to the previous year, for the urban areas, attributable mostly to vehicular emission reductions. For the capital city of Athens, weekly reductions in the TROPOMI tropospheric $NO_2$ columns, between -8% and -43%, for the seven of the eight weeks studied were found, corroborated by the space-born OMI/Aura observations as well as ground-based Multi-Axis Differential Optical Absorption Spectroscopy, MAXDOAS, measurements. Stronger reductions were reported by seven in situ air quality stations in Athens that reported measurements to the European Environmental Agency Air Quality database, with monthly decreases reaching -40% for the month of April 2020. In order to eliminate the expected meteorological effects on the observed $NO_2$ levels, Chemical Transport Modelling simulations, provided by the LOTOS-EUROS CTM, show that the magnitude of these satellite-sensed reductions cannot solely be attributed to the difference in meteorological factors affecting $NO_2$ levels during March and April 2020 and the equivalent time periods of the previous year. Taking this factor into account, the resulting decline due to the COVID-19 related measures was estimated to range between -10% and -20% for the different spatiotemporal scales studied in this work, taking into account possible uncertainties of the methodology considering the low tropospheric $NO_2$ levels observed around Greece.

**Author Contributions:** The data analysis was performed by I.S., A.K. and M.E.K.; methodology and conceptualization by D.S. and I.P.; software development by I.S. and A.K.; writing—original draft preparation by M.E.K.; review and editing by D.S., A.S., A.M., J.v.G. and H.E. All authors have read and agreed to the published version of the manuscript.

**Funding:** This research has been co-financed by the European Union (European Regional Development Fund) and Greek national funds through the Operational Program "Competitiveness, Entrepreneurship and Innovation" (NSRF 2014-2020) by the "Panhellenic Infrastructure for Atmospheric Composition and Climate Change" project (MIS 5021516) and well as the "Innovative system for Air Quality Monitoring and Forecasting" project [code T1EDK-01697, MIS 5031298), implemented under the Action "Reinforcement of the Research and Innovation" Infrastructure.

**Acknowledgments:** We acknowledge the usage of modified Copernicus Sentinel data 2019-2020. Results presented in this work have been produced using the Aristotle University of Thessaloniki (AUTh) High Performance Computing Infrastructure and Resources. M.E.K., I.S. and D.S. would like to acknowledge the support provided by the IT Center of the AUTh throughout the progress of this research work. M.E.K. and A.K. would also like to acknowledge the support provided by the Atmospheric Toolbox®.

**Conflicts of Interest:** The authors declare no conflict of interest.

**Data availability:** The S5P/TROPOMI data are publicly available via the Copernicus Open Data Access Hub, https://scihub.copernicus.eu/. The LOTOS-EUROS simulations are available upon request. The air quality monitoring station data are publicly available via the European Environmental Agency Air Quality monitoring service, https://discomap.eea.europa.eu/map/fme/AirQualityExport.htm and the Greek Ministry Environment & Energy monitoring network, https://ypen.gov.gr/perivallon/poiotita-tis-atmosfairas/dedomena-metriseon-atmosfairikis-rypansis/. The OMI/Aura $NO_2$ cloud-screened tropospheric column L3 global gridded 0.25x0.25° v003 product are publicly available from the NASA EarthData Giovanni repository, https://giovanni.gsfc.nasa.gov/giovanni/. The MAXDOAS observations discussed in this text are publicly available from https://mpc-vdaf-server.tropomi.eu/no2/no2-offl-maxdoas/athens.

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
