# Peer review of "Sudden changes in nitrogen dioxide emissions over Greece due to lockdown after the outbreak of COVID-19 2"

_Atmospheric Chemistry and Physics, 2020_

## Referee Comment (RC1) · Anonymous Referee #1 · 24 Aug 2020

The work of Koukouli et al., provides an insight on the matter of reduction of NO2 during the time of severe COVID-19 related measures at Greece, which is the most discussed topic among scientists this year. Data from TROPOMI and in situ measurements are used to compare the concentrations in March and April of 2019 and 2020. Also, a CTM is used to investigate the differences driven by meteorology during these periods. The subject is very important, and authors have used state of the art data to investigate it. However, I think that are some major improvements needed to be applied before publishing in ACP.

My major concern is the fact that only 2019 data are used to define the drop due to

COVID-19 measures. Do we know that March and April 2019 are representative for the area ? Can we eliminate the possibility of extreme high values during this period? TROPOMI NO2 is not available from earlier periods, but there are retrievals from other satellites. The spatial resolution is poorer and some works have already pinpointed that there are hotspots that were ignored due to larger pixel size. Also, we should know what is the usual variability of NO2 in the area. Are these drops something usual or it is something extremely rare? Hence, some work should be done in order to properly compare the absolute levels and provide to some reference values before defining the drop during the lockdown. Also, the same should be done with in-situ data (where available).

The other major concern is the meteorological variations and how they are used in the model. The CMT model is not described in detail. Reading paragraph 2.2 multiple times, it is not clear what are the inputs of the model , hence, it is impossibleto inter-pretthe results. Also, since the comparisons in the literature are show low correlation coefficient for this model, I think some comparison (for a non-lockdown period) should be presented. Figure 1 is not appropriate for understanding the credibility of the output. Additionally, any information on the variation of key parameters (mainly Boundary Layer, wind speed and Solar Irradiance) during the period would be very useful for understanding the conditions. What was the parameter that drove the theoretical increase in Patra (figure 5 upper plot). Keep in mind that the stricter lockdown period was April 2020, were the drops are lower (even increase is observed at Volos). Thus, the discussion about the meteorological conditions how they affected the NO2 columnar retrievals should be deeper. There are number of findings that cannot be explained, hence question the validity of the approach. Both the approach and the results should be justified thoroughly.

Some specific comments: L58-61 I think here some details should be added about the monthly values, the seasonal cycle and the extreme values of this period. Since, these data are available, I would recommend a timeseries figure at least for Athens

and Thessaloniki.

L102 Some literature should be provided about any possible biases among OMI and TROPOMI retrievals.

L119 It should be noted here, that a lot of industrial activities didn't stop during the lockdown or they were just slow downed. Thus, workers were moving to industrial areas and probably some behaviors could be explained by that (eg Elefsina from in situ measurements).

L.130 The way it is stated gives the impression that TROPOMI retrievals of NO2 commence at August 2019. Please restate to be clear.

L143 ROCRV please explain the abbreviation

L145 This behavior should be considered more. Since during the drop of this period, NO2 values became very similar to background ones, some noise could be added.

2.2 Is there any estimation on the diurnal cycle of NO2 in the area? Is the overpass time of TROPOMI representative? This should also be discussed when selecting the output step of the CTM. Also, please provide details on the input of CTM as asked in general comments.

L208 State what this meteorology includes.

L211 This expectations should be justified properly.

L215 Are there any data or statistics of the clear days during 2020? Also, provide some some information of the actual number of days used (with qa>0.75) for monthly averages. How does cloudy days affect the photochemistry of NO2?

L241 The fact that during the harsher lockdown, NO2 decrease was not as high as in March (when 1/3 of the month was almost with normal activities) should be discussed. Could be a sampling issue?

Figure 2 It seems that another NO2 hotspot is located at western Macedonia, possibly in the coal mining and thermal factories area. Probably it would be interesting to focus also on that (probably compared with the statistics of energy demand provided in 1.2)

Table 2 I recommend to provide also the number of days used for each monthly average retrieval.

L 280-291 I suggest to focus more on these in-situ data. Plots from the supplement could be useful here. Also, it would be useful to calculate also the monthly mean with the days with TROPOMI retrievals in each case and compare directly the differences.

Figure 5 lower plot. These is the main conclusion of the work. Still, if the meteorology variations was properly subtracted, the results for Patra and Herakleion are unexplainable. Further investigation is needed on these cases. Any possibility of an artifact should be eliminated before considering as a valid conclusion. Again, sampling from satellite and meteorological inputs are the main sources of concern for these results.

L311 This is also a crucial finding about the naval activities. I would consider moving the figure to main manuscript instead of the supplement.

Figure 6. How these weeks were selected? Sampling reasons? Please explain explicit. Also, I suggest to use the in-situ data in order to validate the areas that have the larger differences.

Figure 6 Lower plot. Also, the last week have a high increase when subtracting meteorology. Although the end of the lockdown was announced, this is very strange. It could be explained with low values in 2019 (when this was orthodox easter week), but I think some investigation needed also for that.

Please also note the supplement to this comment:
https://acp.copernicus.org/preprints/acp-2020-600/acp-2020-600-RC1-supplement.pdf

---

## Referee Comment (RC2) · Anonymous Referee #2 · 19 Oct 2020

Koukouli et al. present the changes in NO2 pollution levels in six major Greek cities resulting from the COVID-19 public health measures. They use satellite observations from the TROPOMI instrument and model simulations using the LOTOS-EUROS CTM. They find an average decrease in TROPOMI NO2 abundancies of -22% in March 2020 and -11% in April with respect to the 2019 levels over these cities, which is mostly attributed to traffic restrictions. The model simulations for 2019 and 2020 using the same anthropogenic emission inventory can inform about the contribution of emission changes to the observed column. This study adds to an growing body of literature on the effects of COVID-19 on air pollution. It is timely and the first study discussing the effects of COVID-19 measures on air quality in Greek cities. However, the discussion

is generally somewhat limited, and in many instances, the analysis of the results is unconvincing. The article suffers from many typos and language errors and will need a careful rereading before resubmission. I recommend publication after my concerns listed below are adequately addressed in a revised version.

Major comments

1) The decreases along the shipping lanes are interesting but not sufficiently discussed. How does the model perform along the major lanes and how does it compare to the observed decrease? What do shipping activity data tell us about the changes in traffic during the shutdowns? Can you provide a map of the column differences between March 2019 and March 2020 over the sea? From Figure S2, it is clear that a large hot spot located to the east of Athens in March 2019 disappears in March 2020. Is this due to a reduction of ship activity or to meteorological changes? A detailed model-data comparison along ship tracks should be presented. Figure S2 should also be moved to the main manuscript.

2) The comparisons with in situ stations are important for validating your results and should be part of the main text. These comparisons could help to interpret the changes sensed by TROPOMI. Although these comparisons should be taken with caution, it is very important to compare the satellite-based changes with the changes measured locally.

3) I have serious doubts about the derived strong emission decrease (-37%) in Patra (Figure 5). Possible reasons for such a change should be analysed and discussed. Is this change expected based on in situ observations? Another concern is the 26% increase in emissions in Athens during the last week of April (Figure 7). Is this increase due to increased anthropogenic emissions (traffic, energy or industrial sectors) or can this be attributed to modelling uncertainties and/or noise in the data?

4) At the end of Section 2.3, ithe authors claim that that TROPOMI NO2 averaging kernels are not needed because only relative column changes are considered. This

argument is not entirely correct. The averaging kernels could very well be different in 2019 and 2020, and their effects should not be overlooked. I strongly recommend their use in the revised manuscript, or, some demonstration that their effects can be ignored.

5) Sampling problems are not adequately discussed. Some of the model-data discrepancies could be due to sampling issues. For how many days are data available per month and per city? Is this taken into account in the comparisons with the model?

6) The column decreases in March are more important than in April, as turns out from Table 2. This is at odds with the full lock-down period that you mention in Section 1.4 (23 March to 4 May). One would expect the decrease to be more significant in April than in March. How can we explain this? Do we have evidence that the lockdown was not (or less) enforced in April? As a matter of fact, the in situ data (Figure S1) indicate that the decreases are stronger in April than in March. A discussion is warranted. Moreover, there is inconsistency between the values for Heraklion of Table 2 and Figure 4.

Minor comments

There is a recurrent problem with the format of citations. The authors should follow the journal guidelines for literature citations. Footnotes should be removed. Consider adding them in the reference list or insert them in the text (e.g. for URLs). The article needs to be corrected for typos and language mistakes. Consider a careful reading before resubmission.

- l.24: "second largest sector", which is the first one?

- l.31: "we aim to show the quantifiable and beyond doubt decline", rephrase e.g. as: "we aim to quantify the decline"

- l.33: read "hereafter"

- l.36: "we enumerate the improvement", do you mean "we quantify the improvement"?

- l.51: remove "among others"

- l.53: remove "issues"

- Section 1.2: Too many details that are not used afterwards. Consider rewriting to ease readability. In l.73, do you mean "by 2.1%" lower? in l.77, remove 'the work of". In l.80, "If we assume that years 2019 and 2020 were not exceptional": what do you mean by exceptional? In l.82: "will not bare a significant of the emission", something is missing here. Reread carefully.

- l.92: read 'reductions'

- l.95: '25% decrease'

- l.98: update reference

- l.103: Add more references for other regions of the world where lockdown effects on air quality have been studied.

- Section 1.4 is very long and too much detailed. This information is not used later on in the discussion. To interpret the results, one needs to know the beginning and end of the lockdown per city. Was the lockdown nationwide? You could replace this section by a table including this information for easy reference.

- l.111: "coffee shops"

- l.121: remove "discreet".

- l.121-122: The sentence can be removed. In fact, hardly any dates have been used afterwards.

- l.133: read "point"

- Figure 2 is mentioned before Figure 1 in the text. What is the benefit of showing Figure 1?

- Figures 1 and 2 are not easy to read. Can you improve the scale? We cannot even see higher columns in Patra or Heraklion. In Figure 2, could you add additional panels

with the column differences?

- l.169: "distinct", do you mean "specific"?

- l.173: explain what MAX-DOAS stands for

- l.182: This is repeated elsewhere, reread carefully to avoid repetitions.

- l.197: read "wind speed"

- Table 1, Figure 3 and l.249-259 have a lot of repetitions.

- Why do you show the month of March in Figure 3? From Section 1.4 it looks like March was for only half affected by the measures. Wouldn't it be more interesting to show April

- l.312-317: the discussion should be extended, see point 1 above.

---

## Author Comment (AC1) · 29 Dec 2020

We thank Reviewer #1 for offering us the possibility to enrich our article. Point-by-point replies to all comments are given in maroon colour below.

Major comments

1) My major concern is the fact that only 2019 data are used to define the drop due to COVID-19 measures. Do we know that March and April 2019 are representative for the area ? Can we eliminate the possibility of extreme high values during this period? TROPOMI NO2 is not available from earlier periods, but there are retrievals from other satellites. The spatial resolution is poorer and some works have already pinpointed that there are hotspots that were ignored due to larger pixel size. Also, we should know what is the usual variability of NO2 in the area. Are these drops something usual or it is something extremely rare? Hence, some work should be done in order to properly compare the absolute levels and provide to some reference values before defining the drop during the lockdown. Also, the same should be done with in-situ data (where available).

This is of course a very sensible comment which we have tackled by calculating and including in the manuscript climatological means over Athens both from satellite observations, using OMI/Aura since 2005, and in situ observations, using both EEA/Greek Ministry of Environment data since 2005 – for consistency with the satellite observations. These findings were added in Section 3.2, Figures 7 & 8, and discussion therein.

2) The other major concern is the meteorological variations and how they are used in the model. The CMT model is not described in detail. Reading paragraph 2.2 multiple times, it is not clear what are the inputs of the model, hence, it is impossible to interpret the results. Also, since the comparisons in the literature are show low correlation coefficient for this model, I think some comparison (for a non-lockdown period) should be presented. Figure 1 is not appropriate for understanding the credibility of the output. Additionally, any information on the variation of key parameters (mainly Boundary Layer, wind speed and Solar Irradiance) during the period would be very useful for understanding the conditions. What was the parameter that driven the theoretical increase in Patra (figure 5 upper plot). Keep in mind that the stricter lockdown period was April 2020, were the drops are lower (even increase is observed at Volos). Thus, the discussion about the meteorological conditions how they affected the NO2 columnar retrievals should be deeper. There are number of findings that cannot be explained, hence question the validity of the approach. Both the approach and the results should be justified thoroughly.

Since this manuscript was submitted a paper presenting the CTM, the model simulations over Greece for NO$_2$ profiles and columns, as well as extensive comparison/validation against S5P/TROPOMI, MAXDOAS and in situ surface measurements has been published in Skoulidou et al., 2020. We have hence updated Section 2.2, i.e. the presentation of the CTM used in this work, to include findings from the Skoulidou et al. paper, references to specific conclusions, as well as deleting the original Figure 1 which, indeed, did not add much to our document. Furthermore, we have reported in detail all the meteorological parameters that

the CTM ingests from the Operational ECMWF forecast datasets, including the temperature and wind fields for the eight weeks studied over Athens in Figures S5 & S6. We are not aware of any publication that has performed sensitivity studies/validation of how the ECMWF meteorology affects the $NO_2$ output parameters of the LOTOS-EUROS CTM, hence we proceed on faith that the well-established operational ECMWF forecast datasets are of high accuracy and suitable for such air quality studies. Since the methodology chosen in this work to account for the meteorological variability relies of computing differences, and is not based on absolute values, it is justifiably assumed that any systematic error of the CTM ingestion and subsequent analysis based on the meteorological fields, is filtered out. Furthermore, we have included the convolution of the LOTOS-EUROS profiles to the TROPOMI averaging kernels, an operation performed by a LOTOS-EUROS module. As a result, some numerical findings that were indeed surprising have changed without however altering the main take away message of this work. It has indeed been found that over locations with low tropospheric $NO_2$ columns, small loads, within the detectability of the satellite observations may lead to large, unphysical differences. A discussion to that effect had been added in Section 3.1 and the abstract/conclusions were changed accordingly.

Skoulidou, I., Koukouli, M.-E., Manders, A., Segers, A., Karagkiozidis, D., Gratsea, M., Balis, D., Bais, A., Gerasopoulos, E., Stavrakou, T., van Geffen, J., Eskes, H., and Richter, A.: Evaluation of the LOTOS-EUROS NO2 simulations using ground-based measurements and S5P/TROPOMI observations over Greece, Atmos. Chem. Phys. Discuss., https://doi.org/10.5194/acp-2020-987, in review, 2020.

**Some specific comments:**

L58-61 I think here some details should be added about the monthly values, the seasonal cycle and the extreme values of this period. Since, these data are available, I would recommend a timeseries figure at least for Athens and Thessaloniki.

The values reported here are extracted by the official EU/EEA report [EEA, 2019] and were not calculated by the co-author team. The purpose of this section is to introduce the author to the levels of air quality over Greece in a general manner, since most times the air quality over heavily polluted sites is reported in literature, such as Eastern China, US cities and power plants, the Po Valley in Italy, the Benelux area, etc, and scientists are more familiar with those. More detailed analysis on the in situ observations over Athens, including a climatological study and discussion thereof performed by the co-author team, has been included in Section 3.2, Figure 8, as Figures S3 & S4 of the updated supplement.

L102 Some literature should be provided about any possible biases among OMI and TROPOMI retrievals.

In the original version of the article, where no OMI/Aura data were used, it was not deemed important to mention such biases. We have now included an analysis of the OMI/Aura long term observations to examine whether the months studied in this work follow the climatological means or not, see Section 3.2 and Figure 7.

L119 It should be noted here, that a lot of industrial activities didn't stop during the lockdown or they were just slow downed. Thus, workers were moving to industrial areas and probably some behaviors could be explained by that (eg Elefsina from in situ measurements).

We have added this comment in the section.

L.130 The way it is stated gives the impression that TROPOMI retrievals of NO2 commence at August 2019. Please restate to be clear.

Reworded accordingly.

L143 ROCRV please explain the abbreviation

Abbreviation and reference altered as per ACP rules.

L145 This behavior should be considered more. Since during the drop of this period, NO2 values became very similar to background ones, some noise could be added.

This is of course correct and we have added a discussion in the revised text over locations that had low tropospheric NO$_2$ loads to begin with.

2.2 Is there any estimation on the diurnal cycle of NO2 in the area? Is the overpass time of TROPOMI representative? This should also be discussed when selecting the output step of the CTM. Also, please provide details on the input of CTM as asked in general comments.

We have updated this section to include information on the meteorological fields input in the CTM and references to the Skoulidou et al., 2020, paper where the model set up and simulations over Greece are presented and validated. The TROPOMI overpass time over Greece represents the decline in the NO$_2$ created by early morning traffic, while comparisons between TROPOMI and the CTM runs were also included in the aforementioned article. Since the LOTOS-EUROS includes a module which performs the convolution to the satellite AKs, the model finds the closest output time to the satellite observations. This is, for most days, 12:00, while – in case of two orbits covering the domain in the same day – the previous hour may also be chosen. Again, this time is also after the expected NO$_2$ peak due to traffic emissions.

L208 State what this meteorology includes.

The relevant text was added in Section 2.2.

L211 This expectations should be justified properly.

The original phrase "*We cannot of course exclude the possibility that the LOTOS-EUROS model has biases in the resulting NO$_2$ column depending on the meteorological conditions, for example due to uncertainties in mixing under stable conditions,…*" was erroneously inserted after a long discussion within the co-author team on quantifying the possible meteorology-related biases of the LOTOS-EUROS CTM for our Skoulidou et al. work. The statement about mixing under stable conditions refers to the night time biases revealed by the work of Skoulidou et al., 2020, for night time comparisons between in situ and model surface concentrations, where stable conditions are expected.

As we are not aware of a study which examines possible meteorology biases in Lotos-Euros, and based on the high acclaim that the CTM holds by the community, we have to take it on faith that indeed possible meteorology-induced biases for 12:00 UTC cancel out in this methodology which is based on differences and not absolute values.

L215 Are there any data or statistics of the clear days during 2020? Also, provide some some information of the actual number of days used (with qa>0.75) for monthly averages. How does cloudy days affect the photochemistry of NO2?

In this work we have chosen pixels with associated qa > 75, hence permitting observations where a very small percentage of the pixel may be covered by clouds, as recommended by the relevant TROPOMI Product User Manual, https://sentinel.esa.int/documents/247904/2474726/Sentinel-5P-Level-2-Product-User-Manual-Nitrogen-Dioxide. Statistics on the available pixels/days for the monthly mean analysis [Section 3.1] and the weekly analysis over Athens [Section 3.2] have been added. A discussion on cloudiness parameters affecting the photochemistry of NO$_2$ was not included in this text since only one specific hour of the day is discussed for the entire text.

L241 The fact that during the harsher lockdown, NO2 decrease was not as high as in March (when 1/3 of the month was almost with normal activities) should be discussed. Could be a sampling issue?

Indeed the differences observed by TROPOMI in March are stronger than in April. This was curious to us as well, which is why we followed advice and studied the climatological mean 2005-2018 and the deviations of years 2019 and 2020 sensed by the OMI/Aura satellite sensor (see Figure 7 and discussion in Section 3.2). The OMI/Aura observations agree with TROPOMI, in the sense that larger differences are found for the months of March than the months of April between 2019 and 2020. Furthermore, the ground-based Multi-Axis Differential Optical Absorption Spectroscopy, MAXDOAS, station in Athens also reported higher columns in March 2019 than April 2019, which again leads to smaller differences for April 2020 than for March 2020. See Section 6.3.4, page 52, of the official quarterly validation report of the TROPOMI Mission Performance Center, https://mpc-vdaf.tropomi.eu/ProjectDir/reports/pdf/S5P-MPC-IASB-ROCVR-08.01.01-

. These findings are also discussed in our new manuscript in Sections 2.1 & 3.2.

Figure 2 It seems that another NO2 hotspot is located at western Macedonia, possibly in the coal mining and thermal factories area. Probably it would be interesting to focus also on that (probably compared with the statistics of energy demand provided in 1.2)

Indeed, near the Greek border where the main lignite-burning power plant complex of Ptolemaida is located there exists a similar power plant in Northern Macedonia, in Novaci. We often observe outflow and inter-regional transport of pollutants in the area, bringing higher overall pollutant loads due to this geographical proximity of the sources and the prevailing winds and topography of the region.

Table 2 I recommend to provide also the number of days used for each monthly average retrieval.

We have added the statistics for the available pixels in the text discussing the findings of the monthly analysis.

L 280-291 I suggest to focus more on these in-situ data. Plots from the supplement could be useful here. Also, it would be useful to calculate also the monthly mean with the days with TROPOMI retrievals in each case and compare directly the differences.

Unfortunately, comparing in-situ data [i.e. surface concentrations] directly to satellite tropospheric columnar assessments is not an easy task, nor does it guarantee meaningful results, as you are aware I am sure. We have however included a discussion on what the in situ observations show, both from the climatology aspect, as well as for the lockdown months in question in Section 3.2 Indeed, there appears to be a disagreement between the magnitudes of the decreases between in situ and space-born observations, both from OMI and TROPOMI. We can only postulate that the air quality stations are much more sensitive to instantaneous changes in traffic conditions on the point locations they are situated, which the space-born sensors with their larger field of view cannot resolve.

Figure 5 lower plot. These is the main conclusion of the work. Still, if the meteorology variations was properly subtracted, the results for Patra and Herakleion are unexplainable. Further investigation is needed on these cases. Any possibility of an artifact should be eliminated before considering as a valid conclusion. Again, sampling from satellite and meteorological inputs are the main sources of concern for these results.

From this analysis it became apparent that for the cases of low monthly mean levels, such as those over the smaller Greek cities, the variability within the standard deviation introduced large absolute and relative differences between the satellite 2020/2019 difference and the CTM 2020/2019 difference. As a result we have added a discussion in Section 3.1, as well as altered the abstract & conclusions.

L311 This is also a crucial finding about the naval activities. I would consider moving the figure to main manuscript instead of the supplement.

Since we composed this article, we have worked separately on the topic of shipping emissions and we indeed realized that there are a lot of interesting findings that can be extracted for this activity. We have hence decided to remove all shipping references from this article in favor of the autonomous work on the subject of shipping activities we are currently preparing for publication.

Figure 6. How these weeks were selected? Sampling reasons? Please explain explicit. Also, I suggest to use the in-situ data in order to validate the areas that have the larger differences.

For Figure 6 [now Figure 5] we wished to show, in a pictorial way, the variability of the $NO_2$ across some of the weeks studied and we picked the weeks with the highest possible amount of observations across the pixels studied for the case of Athens. We have added a discussion on what the in situ stations show in Section 3.2 as well as Figure 8, and Figures S3 & S4 in the supplement.

Figure 6 Lower plot. Also, the last week have a high increase when subtracting meteorology. Although the end of the lockdown was announced, this is very strange. It could be explained with low values in 2019 (when this was orthodox Easter week), but I think some investigation needed also for that.

Even though we have every confidence that the meteorology is well taken care of by LOTOS-EUROS, we have added the weekly meteorological mean vector wind directions and speeds, as well as the surface temperature, in Figures S5 & S6 in the supplement. The mean vector wind speed and direction, overlain as arrows in Figure S5, is very similar with mostly predominant northern winds and very few cases of southerly winds. In the equivalent rose diagrams, Figure S6, we note again that the main wind directions appear similar between the two periods [2019 in the left and 2020 in the right column] apart from the last week of April [bottom row] were indeed the two weeks had very different directions, for the same magnitude.

---

## Author Comment (AC2) · 29 Dec 2020

We thank Reviewer #2 for offering us the possibility to enrich our article. Point-by-point replies to all comments are given in maroon colour below.

Major comments

1)      The decreases along the shipping lanes are interesting but not sufficiently discussed. How does the model perform along the major lanes and how does it compare to the observed decrease? What do shipping activity data tell us about the changes in traffic during the shutdowns? Can you provide a map of the column differences between March 2019 and March 2020 over the sea? From Figure S2, it is clear that a large hot spot located to the east of Athens in March 2019 disappears in March 2020. Is this due to a reduction of ship activity or to meteorological changes? A detailed model-data comparison along ship tracks should be presented. Figure S2 should also be moved to the main manuscript.

Since we composed this article, we have worked separately on the topic of shipping emissions and we indeed realized that there are a lot of interesting findings that can be extracted for this activity. We have hence decided to remove all shipping references from this article in favour of the autonomous work on the subject of shipping activities we are currently preparing for publication.

2)      The comparisons with in situ stations are important for validating your results and should be part of the main text. These comparisons could help to interpret the changes sensed by TROPOMI. Although these comparisons should be taken with caution, it is very important to compare the satellite-based changes with the changes measured locally.

We have enriched our work by showing a more comprehensive analysis of what air quality stations around Athens have reported, both in the main text (Figure 8 and discussion thereof) as well as in the supplementary material (Figures S3 & S4.) Furthermore, our work in comparing the LOTOS-ERUROS simulations to TROPOMI, MAXDOAS and in situ stations around Greece has been published n Skoulidou et al., 2020, and references/discussions to the validation/comparison of the model runs used also in this work have been added to the flow of this text (see Section 2.2)

Skoulidou, I., Koukouli, M.-E., Manders, A., Segers, A., Karagkiozidis, D., Gratsea, M., Balis, D., Bais, A., Gerasopoulos, E., Stavrakou, T., van Geffen, J., Eskes, H., and Richter, A.: Evaluation of the LOTOS-EUROS NO2 simulations using ground-based measurements and S5P/TROPOMI observations over Greece, Atmos. Chem. Phys. Discuss., https://doi.org/10.5194/acp-2020-987, in review, 2020.

3)      I have serious doubts about the derived strong emission decrease (-37%) in Patra (Figure 5). Possible reasons for such a change should be analysed and discussed. Is this change expected based on in situ observations? Another concern is the 26% increase in emissions in Athens during the last week of April (Figure 7). Is this increase due to increased

anthropogenic emissions (traffic, energy or industrial sectors) or can this be attributed to modelling uncertainties and/or noise in the data?

Following suggestions by both referees, we have updated our model runs using the convolution of the satellite averaging kernels for the production of the columnar $NO_2$ estimates by the CTM. As expected, some numerical results are now different [such as the high negative decrease you mention] while the main findings, such as the relative levels and reductions, were not altered. It became apparent that for the cases of low monthly mean levels, such as those over the smaller Greek cities, the variability within the standard deviation introduced large absolute and relative differences between the satellite 2020/2019 difference and the CTM 2020/2019 difference. As a result we have added a discussion in Section 3.1, as well as altered the abstract & conclusions.

4)        At the end of Section 2.3, ithe authors claim that that TROPOMI NO2 averaging kernels are not needed because only relative column changes are considered. This argument is not entirely correct. The averaging kernels could very well be different in 2019 and 2020, and their effects should not be overlooked. I strongly recommend their use in the revised manuscript, or, some demonstration that their effects can be ignored.

In order to be absolutely certain that not using of the AKs does not introduce differences in our main findings, we have repeated the entire analysis convolving the CTM profiles to the TROPOMI AKs. While for the locations with the smaller $NO_2$ loads the convolution did bring some differences in the resulting percentage values, the main findings remain and are indeed strengthened in this manner.

5) Sampling problems are not adequately discussed. Some of the model-data discrepancies could be due to sampling issues. For how many days are data available per month and per city? Is this taken into account in the comparisons with the model?

All findings in this work depend on the collocation of the satellite dataset to the CTM results, hence the exact same pixels are used for both datasets and this type of sampling issue is not expected. Sampling might be an issue if between years the number of collocations varies dramatically. The difference in number of available pixels for the monthly mean comparisons are added to Section 3.1 and the number of available days for the case of the weekly analysis over Athens was added to Section 3.2 .

6) The column decreases in March are more important than in April, as turns out from Table 2. This is at odds with the full lock-down period that you mention in Section 1.4 (23 March to 4 May). One would expect the decrease to be more significant in April than in March. How can we explain this? Do we have evidence that the lockdown was not (or less) enforced in April? As a matter of fact, the in situ data (Figure S1) indicate that the decreases are stronger in April than in March. A discussion is warranted. Moreover, there is inconsistency between the values for Heraklion of Table 2 and Figure 4.

Finally, thank you for spotting the differences for the mean levels of Heraklion between table and Figure, it led us to note that the calculation for Patras was also not performed properly. This is simply due to the fact that we were examining different pixels as representative for each of these locations and this led to the errors in the numerics of the table.

As to the well spotted point that the differences in March are stronger than in April. This was indeed curious to us as well, which is why we followed advice and studied the climatological mean 2005-2018 and the deviations of years 2019 and 2020 sensed by the OMI/Aura satellite sensor (see Figure 7 and discussion in Section 3.2). The OMI/Aura observations agree with TROPOMI, in that larger differences are found for the months of March than the months of April between 2019 and 2020. Furthermore, the ground-based Multi-Axis Differential Optical Absorption Spectroscopy, MAXDOAS, station in Athens also reported higher columns in March 2019 than April 2019, which again leads to smaller differences for April 2020 than for March 2020. See Section 6.3.4, page 52, of the official quarterly validation report of the TROPOMI Mission Performance Center, https://mpc-vdaf.tropomi.eu/ProjectDir/reports/pdf/S5P-MPC-IASB-ROCVR-08.01.01-

20200921_FINAL.pdf. These findings are also discussed in our new manuscript in Sections 2.1 & 3.2.

Minor comments

There is a recurrent problem with the format of citations. The authors should follow the journal guidelines for literature citations. Footnotes should be removed. Consider adding them in the reference list or insert them in the text (e.g. for URLs). The article needs to be corrected for typos and language mistakes. Consider a careful reading before resubmission.

We have removed all references previously given as footnotes to match the ACP guidelines and re-read the article to weed out typos and language mistakes.

- l.24: "second largest sector", which is the first one?

The first one is industry, we have added the clarification in the abstract.

- l.31: "we aim to show the quantifiable and beyond doubt decline", rephrase e.g. as: "we aim to quantify the decline"

Text altered as suggested.

- l.33: read "hereafter"

Text altered as suggested.

- l.36: "we enumerate the improvement", do you mean "we quantify the improvement"?

Text altered as suggested.

- l.51: remove "among others"

Text altered as suggested.

- l.53: remove "issues"

Text altered as suggested.

- Section 1.2: Too many details that are not used afterwards. Consider rewriting to ease readability.

As requested by yourself later on in this review, and the fact the more publications have appeared since we composed this section, we added a paragraph of reported findings in the end of this Section.

- In l.73, do you mean "by 2.1%" lower?

  Yes, the minus sign was getting left behind in the previous line, text altered as suggested.

- in l.77, remove 'the work of".

  Text altered as suggested.

- In l.80, "If we assume that years 2019 and 2020 were not exceptional": what do you mean by exceptional?
- In l.82: "will not bare a significant of the emission", something is missing here. Reread carefully.

  [both points above refer to the same sentence]

*True, well spotted. Phrase now reads:* If we assume that years 2019 and 2020 were not exceptional in their temperature levels for the spring months, then it follows that changes in central heating emissions will not be bare a significant part of the emission changes observed.

- l.92: read 'reductions'

Text altered as suggested.

- l.95: '25% decrease'

Text altered as suggested.

- l.98: update reference

Reference updated.

- l.103: Add more references for other regions of the world where lockdown effects on air quality have been studied.

References added as suggested.

- Section 1.4 is very long and too much detailed. This information is not used later on in the discussion. To interpret the results, one needs to know the beginning and end of the lockdown per city. Was the lockdown nationwide? You could replace this section by a table including this information for easy reference.

The details provided in this section aimed to show that during March, and well before the total lockdown of March 23rd, numerous sectors of normal life were being shut down one by one which resulted in the anticipated restriction of movement and hence possibly lower exhaust fumes.

- l.111: "coffee shops"

Text altered as suggested.

- l.121: remove "discreet".

Text altered as suggested.

- l.121-122: The sentence can be removed. In fact, hardly any dates have been used afterwards.

Text altered as suggested.

- l.133: read "point"

Text altered as suggested.

- Figure 2 is mentioned before Figure 1 in the text. What is the benefit of showing Figure 1?

Since we submitted this article before the article of Skoulidou et al., 2020, which presents the LOTOS-EUROS set up and first results over the region, we thought the add a figure for demonstrational purposes indeed. Since the Skoulidou et al., 2020, article is now online we have deleted the figure and added model information from that work.

- Figures 1 and 2 are not easy to read. Can you improve the scale? We cannot even see higher columns in Patra or Heraklion.

I tried a number of colour tables/colour scales but it was not possible to both view the hotspots that rise above 8-10x$10^{15}$ molec/cm$^2$ and at the same times cities with NO$_2$ loads between 1-2 10x$10^{15}$ molec/cm$^2$ around the big domain [new Figure 1]. This is why in [new] Figure 2 [zoom-in figures around the cities] the colour bars are not the same for all locations.

- In Figure 2, could you add additional panels with the column differences?

Differences panel added as suggested.

– l.169: "distinct", do you mean "specific"?

Text altered as suggested.

- l.173: explain what MAX-DOAS stands for

Acronym added.

- l.182: This is repeated elsewhere, reread carefully to avoid repetitions.

Indeed, well spotted.

- l.197: read "wind speed"

Text altered as suggested.

- Table 1, Figure 3 and l.249-259 have a lot of repetitions.

We have moved the (previous) Table 1 to the supplement, as TableS1, and re-wrote parts of this section.

- Why do you show the month of March in Figure 3? From Section 1.4 it looks like March was for only half affected by the measures. Wouldn't it be more interesting to show April

As discussed in the text, the NO$_2$ levels reported by TROPOMI over the entire country in April 2019 were lower than in March 2020, seen also by OMI/Aura and the ground-based MAXDOAS of Athens. This fact hinders the methodology chosen for this analysis, for the locations with already low tropospheric NO$_2$ load. We have added the April maps in the supplement, as Figure S1, where the decline for April 2020 is of course still evident.

- l.312-317: the discussion should be extended, see point 1 above.

See discussion under point #1 above.